# Between hope and reality: treatment of genetic diseases through nucleic acid-based drugs
Virginie Baylot ✉, Thi Khanh Le, David Taïeb, Palma Rocchi ✉ & Laurence Colleaux

Rare diseases (RD) affect a small number of people compared to the general population and are mostly genetic in origin. The first clinical signs often appear at birth or in childhood, and patients endure high levels of pain and progressive loss of autonomy frequently associated with short life expectancy. Until recently, the low prevalence of RD and the gatekeeping delay in their diagnosis have long hampered research. The era of nucleic acid (NA)-based therapies has revolutionized the landscape of RD treatment and new hopes arise with the perspectives of disease-modifying drugs development as some NA-based therapies are now entering the clinical stage. Herein, we review NA-based drugs that were approved and are currently under investigation for the treatment of RD. We also discuss the recent structural improvements of NA-based therapeutics and delivery system, which overcome the main limitations in their market expansion and the current approaches that are developed to address the endosomal escape issue. We finally open the discussion on the ethical and societal issues that raise this new technology in terms of regulatory approval and sustainability of production.

Rare diseases (RD), also referred to as orphan diseases, are defined by the European Commission as pathologies that affect less than 5 patients per 10,000 members of the population[1]. There are about 7–8000 known and registered RD (5 new RD are described in the medical literature every month) affecting approximately 350 million people worldwide[2,3]. Most patients with RD suffer of serious, chronic, progressive illnesses. Over 75% of RD appear during childhood, such as proximal spinal muscular atrophy, neurofibromatosis, osteogenesis imperfecta, chondrodysplasia or Rett syndrome, and are often associated with lifelong suffering, impaired quality of life and reduced lifespans[4,5]. It is estimated that a third of patients with RD die before their 5th birthday. Even if the potential causes of RD are not fully understood, 80% are genetic in origin[6]. Genetic RD are driven by a deficient gene that can be inherited or arise from de novo mutations. Due to the limited patient population, RD are neglected pathologies and effective therapies are still not available for more than 95% of the patients suffering from these conditions[7]. The development of new therapeutic tools to treat orphan diseases are often not considered profitable and too expensive, representing a significant unmet medical need for the patients[8].

In 1983, the first attempt to promote orphan drug development was established with the Orphan Drug Act (ODA), which provided financial incentives for the development of new treatments for RD, such as specific financial supports, tax credits, priority review and market[9–11]. The impact of the ODA has been important and hundreds of new orphan drugs were developed. Unfortunately, most of these therapeutic approaches act as symptomatic treatments that delay disease's progression and/or reduce the disorder's impact on the patient's life[5,12]. Existing treatments are mostly based on small molecules and protein-targeting compounds that often display limited efficacy due to low target affinity, inefficient cell penetration and short half-life, leading to drug resistance[13]. Nucleic acid (NA)-based drugs may provide better therapeutic options that aim to fix the genetic problem at its source by modifying and repairing the disease-causing gene[14].

Over the past decades, NA drugs including DNA- and RNA-based therapeutics, have been widely exploited to treat a wide range of diseases, especially genetic disorders, and cancers. The first attempt to deliver a gene coding for resistance to neomycin into lymphocytes harvested from cancer patients was published 30 years ago[15]. After decades of promises, tempered by great deal of failures, NA-based therapies targeting the gene(s) responsible(s) for various human diseases have now become promising therapeutic tools. Remarkable progresses in NA drugs development were made recently as they are, compared to conventional medicines, relatively rapid to develop more cost-effective, and more specific. They include the development of efficient and safe viral-based gene transfer systems, the discovery of short interfering RNA (siRNA), the development of RNA analogs or major advance regarding the bioavailability of NA-based drugs by the introduction of chemical modifications that increase nuclease resistance of DNA and RNA oligonucleotides[14].

Aix Marseille Univ, CNRS, CINAM, ERL INSERM U 1326, CERIMED, Marseille, France. ✉e-mail: virginie.baylot@inserm.fr; palma.rocchi@inserm.fr

**Fig. 1 | The main principal of gene replacement therapy using a viral vector.** Binding of Adeno-associated virus (AAV)-based vectors to specific membrane receptors (1) induces cellular internalization through endocytosis (2). Once inside the host cells, the viral vectors are released from the endosome (3) and shuttled into the nucleus, where the ssDNA is released (4) and undergoes second-strand synthesis to form double-stranded DNA. Subsequently, the transgene is transcribed into corresponding mRNAs (5) that will be translocated to the cytoplasm to generate therapeutic proteins (6). Created with BioRender.com (Agrement number : YC26MKZXYB).

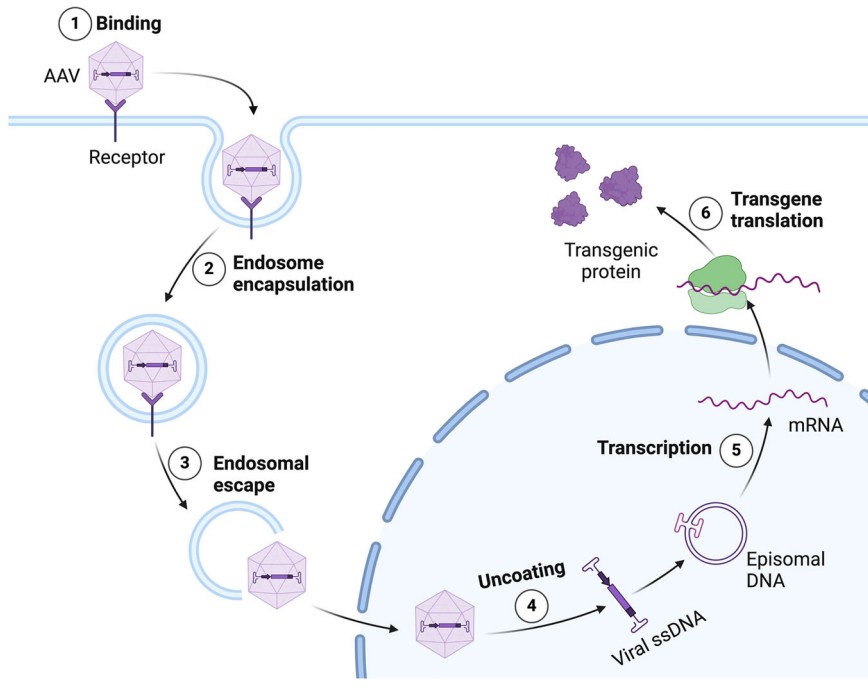

While no NA-based drugs have been approved so far for the treatment of cancer, 23 of them were approved by the U.S. Food and Drug Administration (FDA) and/or the European Medicines Agency (EMA) and/or the Medicines and Healthcare products Regulatory Agency (MHRA) and/or the Japanese Ministry of Health, Labour and Welfare (MHLW) to treat various rare diseases. This review provides an overview of these approved therapies, first describing the rationale underlying the drug design. Then, we address the issue of the delivery system. Finally, the challenges and opportunities raised by this new class of medication are discussed from multiple perspectives.

## Disease-modifying drugs for rare genetic disorders
### Rationale underlying nucleic acid-based drugs
Nucleic acid therapeutics open almost infinite possibilities. They can be designed to i) restore the normal biological functions of deficient proteins; ii) halt the production of abnormal deleterious protein while leaving all other proteins unaffected; iii) increase the synthesis of a protein of interest; iv) hide non-sense mutation on the mRNA that prevent the production of a complete protein or correct mutations that affect mRNA splicing. Depending on the chosen objective, several strategies have been developed.

### Gene replacement therapies
Gene replacement therapy aims to compensate for abnormal or non-functional genes by delivering genetic materials into cells, thus introducing a healthy copy of the gene[16]. The therapeutic gene expression cassette is delivered with the aid of a vector, most commonly a retrovirus or rarely adenovirus, and is typically composed of a promoter that drives gene transcription, the transgene of interest, and a termination signal to end gene transcription[17]. Once being delivered into the host cells, the therapeutic gene is transcribed into mRNA transcripts that are translated into the proteins of interest (Fig. 1). While this replacement is usually partial, producing a small amount of normal gene product is often sufficient to provide significant health benefits. Although the concept was proposed three decades ago, significant advances were only made over the past decade.

Two distinct strategies are used for gene replacement. Genetic material can be delivered to the target cells either ex vivo (when genetic manipulation of cells is undertaken remotely outside the body) or directly in vivo by local or systemic injections[18]. In vivo gene transfer methods deliver directly into patients usually rely on nonintegrating vectors, while ex vivo approaches use integrating vectors to genetically modify explanted stem cells that are reinfused to the same patient[19]. Four treatments are now approved, i.e., Luxturna™, Zolgensma™, Strimvelis™, and Lyfgenia™ while dozens of other treatments are under clinical trials (Table 1).

Approval of gene therapy for Adenosine Deaminase (ADA) deficient severe combined immunodeficiency (SCID) occurred 25 years after the first gene therapy attempt to treat children affected by this disorder, also known as bubble babies[20]. ADA-SCID was an ideal candidate for ex vivo gene transfer in hematopoietic stem cells for several reasons. It is an ubiquitously expressed housekeeping gene, with no need of fine regulation and *ADA* cDNA is small (1.5 kb) and can thus be easily cloned and expressed by lentiviral vectors[21]. Second, relatively low enzymatic levels allow normal immune functions in healthy individuals carrying a normal *ADA* gene. Third, normal or gene-corrected cells have a strong selective survival advantage over *ADA*-deficient cells[21]. After pre-clinical studies demonstrated that gene therapy was feasible and had an acceptable safety profile[22,23], Strimvelis™ (GSK) was the first ex vivo stem cell gene therapy approved by the EMA In May 2016[24]. Three years later, EMA approved Zynteglo™, a gene therapy designed to treat the blood disorder beta-thalassemia launched by Bluebird bio and a first patient was dosed with Zynteglo® in Germany in February 2021. In August 2022, the FDA granted approval to Zynteglo™, marking it as the pioneering cell-based gene therapy for beta-thalassemia patients who need regular red blood cell transfusions. In two phase 3 trials (NCT02906202, NCT03207009) approximately 89% (32/36) of patients receiving Zynteglo™ attained transfusion independence, with a median total hemoglobin level of 11.5 g/dL[25,26]. Yet, marketing is currently suspended due to serious adverse effects (leukemia and myelodysplastic syndrome) observed in patients treated with a similar drug using the same viral vector in a different indication, and failure to reach a reimbursement deal with the German authorities[27].

Due to the ease of access to the diseased tissue, gene therapy may be particularly promising for monogenic skin disorders and researchers have been working on gene therapies for epidermolysis bullosa for over a decade[28]. As a result, in May 2023, the FDA approved, *via* its fast-track process, Vyjuvek™ (Beremagene geperpavec-svdt, also known as B-VEC; Krystal Biotech) a topical gel therapy containing a replication defective herpes-simplex virus type 1 (HSV-1) vector encoding *COL7A1* transgene for the treatment of recessive dystrophic epidermolysis bullosa (Table 1)[29–32].

## Table 1 | Approved gene replacement therapies for genetic diseases

| Trade name (Drug Name) | Strategy | Approval year | Disease | Targeted genes | Vector | Company |
|---|---|---|---|---|---|---|
| Glybera (alipogene tiparvovec) | Ex vivo | 2012[↑2017] | Familial lipoprotein lipase deficiency | lipoprotein lipase (*LPL S447X*) | Adeno Associate Viru 1 (AAV1) | uniQure |
| Strimvelis (GSK2696273) | Ex vivo | 2016 | Severe combined immunodeficiency due to adenosine deaminase deficiency | Adenosine deaminase (*ADA*) | Gamma-retrovirus | GSK |
| Luxturna™ (Voretigene Neparovvec-rzyl) | In vivo | 2017 | RPE65-linked Retinal dystrophy | *RPE65* | Adeno Associate Virus 2 (AAV2) | Spark Therapeutics |
| Zynteglo™ (Betibeglogene autotemcel) | Ex vivo | 2019[suspended] | β-thalassemia | βA-T87Q-globin (modified *β-globin gene*) | Lentivirus (BB305) | Bluebird bio |
| Zolgensma™ (Onasemnogene Abeparvovec-xiol) | In vivo | 2019 | Spinal Muscular atrophy | *SMN1* | Adeno Associate Virus 9 (AAV9) | Novartis |
| Vyjuvek™ (beremagene geperpavec) | In vivo | 2023 | Epidermolysis bullosa | *COL7A1 gene* | Herpes Simplex Virus (HSV-1) | Krystal Biotech |
| Lyfgenia™ (Lovotibeglogene autotemcel) | Ex vivo | 2023 | Sickle Cell Disease | Hemoglobin ($Hb^{AT87Q}$) | Lentivirus (BB305) | Bluebird Bio |

[a]Withdrawal of approval.

Strategies using ex vivo gene editing undoubtedly have a bright future for genetic disorders related to blood cell defects such as hematological disorders or primary immunodeficiency diseases. In contrast, for the vast majority of genetic diseases, the drug must be delivered to the target cells in vivo. Leber congenital amaurosis (LCA) refers to a group of diseases that cause severe vision loss in infancy due to abnormal function and later degeneration of the retina. The first in vivo gene replacement therapy for an inherited disorder in the US, Luxturna® (voretigene neparvovec-rzyl, Spark Therapeutics), was approved by the FDA in 2017 for the treatment of inherited retinal dystrophy caused by biallelic *RPE65* gene mutation (LCA2)[33]. This therapy, that enters a phase I trial (NCT00516477) in 2007, is based on live nonreplicating genetically modified AAV2 capsid that expresses *RPE65*, delivered by a subretinal injection[34]. After demonstration of an acceptable safety profile Luxturna® entered a phase I/II (NCT01208389) in 2010 that further confirmed the safety and efficacy profiles and a durability of the treatment for at least three years. Neither serious adverse events nor deleterious immune responses were observed likely due to the immune privileged route of administration and low viral dose injected[35]. In 2012, Luxturna® entered phase III (NCT00999609) and began to be administered into the contralateral eye of 31 patients (21 patients treated with Luxturna and 10 with the placebo)[36]. The results showed that 13 of the 21 patients (62%) treated with Luxturna™ passed the mobility test at the lowest light level of 1 lux (corresponding to conditions of an inadequately illuminated area at night), while none of the control patients were able to do so. The improvement in patients' vision was maintained for at least three years. In addition, all safety, immune response, and activation of the visual cortex supported the usability of this drug in *RPE65*-associated LCA patients and made Luxturna® the first FDA-approved gene therapy for a genetic disease[37].

Spinal muscular atrophy (SMA) is a devastating autosomal recessive and, typically, childhood-onset neuromuscular disorder caused by loss-of-function mutations in the survival of motor neuron 1 (*SMN1*) gene. *SMN1* mutations result in SMN protein deficiency and cause motor neuron degeneration leading to debilitating and often fatal muscle weakness[38]. In SMA murine models, the administration of a «self-complementary» adenovirus serotype 9-associated virus (scAAV9) vector containing one or more copie(s) of the *SMN* transgene (scAAV9-SMN) showed its ability to cross the blood brain barrier and achieve high levels of neuronal transduction[39–41]. As AAV-based vector genomes can integrate into chromatin, it has been demonstrated that a single intravenous injection of the scAAV9-SMN Zolgensma™ (onasemnogene abeparvovec, Novartis) was sufficient to induce long-lasting therapeutic efficiency[42]. The SMA gene-therapy phase 1 trial named START (NCT02122952) results exceeded all expectations as all 15 SMA infants were alive and free of permanent ventilation, compared with a survival rate of 8% in a historical cohort[43]. Based on these results, Zolgensma® received, in May 2019, the FDA approval as the first-ever systemically delivered AAV gene therapy. Two phase 3 clinical trials STR1VE-US ($n = 22$) and STR1VE-EUROPE ($n = 33$) (NCT03306277, NCT03505099) showed safety and efficacy of commercial-grade Zolgensma®, while a recent study confirmed the widespread biodistribution of vector genomes and transgenes throughout the central nervous system and peripheral organs[44–46].

Based on recent successes and the scientific enthusiasm in gene therapy, the FDA has approved on December 8th 2023, Lyfgenia™ (lovotibeglogene autotemcel - lovo-cel), a lentiviral vector-based gene therapy (Bluebird Bio) for the treatment of patients with sickle cell disease (SCD) ages 12 or older with a history of vaso-occlusive events[47,48]. With beta-thalassemia, SCD is one of the most common monogenic hematological disorders worldwide[49,50]. This pathology is caused by mutations in the hemoglobin β subunit gene (HBB) resulting in ineffective erythropoiesis and severe anemia[51,52]. Patients with SCD need long-life blood transfusion, and available treatment options fail to address the underlying cause of the diseases[53,54]. Lovo-cel (previously named LentiGlobin) is an ex vivo gene therapy that aims the addition of a functional gene that enables production of $Hb^{AT87Q}$ which functions similarly to adult hemoglobin A. The FDA

**Fig. 2 | siRNAs mechanism of action.** After the internalization of nanoparticles (1), siRNAs are released from the endosome (2) and loaded into the RNA-induced silencing complex (RISC) (3). The siRNAs are then incorporated into RISC, leading to the cleavage of the sense strand (4). The antisense strand (guide strand) binds to its target mRNA (5), and the siRNA guide strand bound to RNA induces an endonucleolytic cleavage mediated by the protein Argonaute-2 (Ago2), which prevents mRNA translation (6). Created with BioRender.com (Agreement number: ZF26ML06W7).

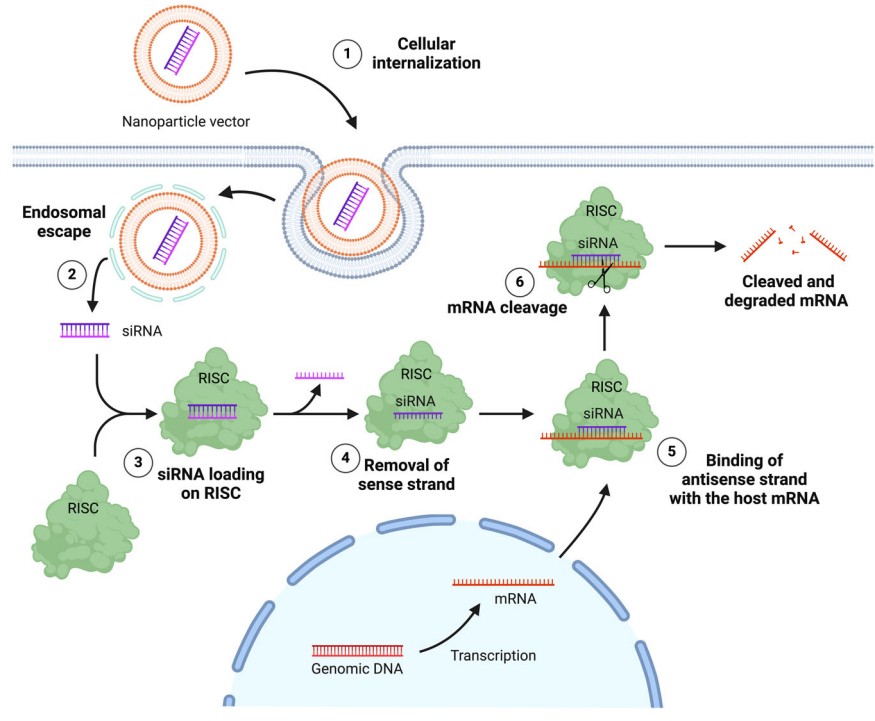

approval of Lyfgenia™ is based on data from patients from the Phase 1/2 HGB-206 study (NCT02140554), notably, the efficacy of this drug was investigated in the HGB-206 Group C trial that enrolled 43 patients[55]. In this cohort, 35 patients (including 8 adolescent patients) received the Lyfgenia™ infusion, in 94% of the patients severe vaso-occlusive crisis were resolved, in the 6-18 months post-infusion, compared with a median of 3.5 events per year without treatment[55]. Among them, 88.2% of the patients experienced no vaso-occlusive event at all.

There are currently hundreds of clinical trials worldwide assessing the therapeutic benefits of gene replacement therapies in many different inherited disorders such as metabolic conditions (Friedreich's Ataxia, Wilson disease), cystic fibrosis or hematological conditions[18].

## RNA therapeutics

The most clinically advanced RNA therapeutics for RD treatments rely on the use of siRNA mediating RNA interference (RNAi) and antisense oligonucleotides (ASO). Both molecules modulate gene expression by binding to the target RNA through Watson-Crick base pairing.

*Small interfering ribonucleic acid (siRNA):* Since 1958, the central dogma stated that DNA was transcribed into RNA and that RNA (called messenger RNA-mRNA) was translated into protein. The discovery of RNA interference (RNAi) in the early 1990's revolutionized the modern biology[56,57]. RNAi refers to a biological process in which double stranded RNAs (dsRNA) could repress gene expression at the post-transcriptional level *via* the degradation of the corresponding mRNA molecules[58]. In 2001, short double-stranded, non-coding RNA molecules (19–25 base pairs long) were reported to mediate RNAi and regulate the expression of genes in mammalian cells[59,60]. These Small interfering RNAs (siRNAs) opened new therapeutic avenues to treat numerous diseases. After their internalization, siRNAs are loaded into the RNA-induced silencing complex (RISC) where it will initiate gene silencing. The incorporation of siRNAs into RISC leads to the cleavage of the sense strand and the binding of the antisense strand (guide strand) to its target mRNA. The siRNA guide strand bounding to RNA induces an endonucleolytic cleavage mediated by the protein Argonaute-2 (Ago2), preventing mRNA translation[61] (Fig. 2).

Challenges faced by siRNA-based therapeutic approaches include off-target effects, efficacy, delivery, and immune system activation. These limitations have been partially overcome by using chemical modifications, ligand conjugation or encapsulation of siRNAs in lipid nanoparticles, leading to the approval of three siRNAs in the context of genetic diseases while other drug candidates are currently in phase 3 clinical trials[62] (Table 2).

Hereditary transthyretin amyloidosis (hATTR amyloidosis) is a fatal autosomal dominant genetic condition caused by mutations in the gene encoding transthyretin (TTR), a liver protein involved in the transport of thyroxine and retinol[63]. The gene mutations cause TTR protein instability leading to its dissociation and the accumulation of toxic and insoluble aggregates of amyloid fibrils in multiple sites including the heart, eyes, peripheral nerves, and kidneys. Patients with this condition develop multiorgan dysfunction and usually die within 5–15 years after diagnosis[63,64]. ONPATTRO® (Patisiran developed by Alnylam Pharmaceuticals) is a siRNA designed to reduce hepatic production of both wild-type and mutant TTR proteins, formulated as a lipid nanoparticle (LNP). A pivotal trial evaluating the safety and efficacy of ONPATTRO® demonstrated its ability to reduce efficiently and rapidly TTR protein levels in patients and improve significantly the neurological manifestations of the disease as well as the patients' quality of life. ONPATTRO® was approved by the EMA and FDA in 2018 for the treatment of adult patients with hATTR amyloidosis with stage 1 or stage 2 polyneuropathy[65,66]. The APOLLO phase 3 trial (NCT01960348) confirmed these previous study results on 170 patients (patisiran, $n = 148$ and placebo, $n = 22$), and demonstrated that ONPATTRO® treatment halts or reverses neuropathy manifestations in 56% of the patients (based on Composite Autonomic Symptom Score-31 (COMPASS-31) questionnaire) and leads to significant benefit across all measure of quality of life (based on the Norfolk Quality of Life-Diabetic Neuropathy (Norfolk QOL-DN) questionnaire) compared with placebo[67,68].

In June 2022, Alnylam® Pharmaceuticals announced that another siRNA that reduces TTR protein production, AMVUTTRA™ (vutrisiran), received the FDA approval, followed by the EMA approval in September 2022, for the treatment of hATTR amyloidosis with polyneuropathy[69]. AMVUTTRA™ is a siRNA covalently linked with three N-acetylgalactosamine sugar molecules, that directly targets wild-type and mutant TTR mRNA. GalNAc molecules recognize and bind to the Asialoglycoprotein (ASGP), a receptor abundantly present at the surface of hepatocytes[70]. Covalent conjugation of siRNA sequences to GalNAc enhances their ability to internalized into cells. Indeed, GalNAc molecules

**Table 2 | Approved siRNA-based therapies for genetic diseases**

| Trade name (Drug Name) | Approval year | Disease | Targeted genes | Vector - modification | Company |
|---|---|---|---|---|---|
| Onpattro™ (Patisiran) | 2018 | Adult patients with hereditary transthyretin mediated amyloidosis | Transthyretin (TTR) | Lipid nanoparticle | Alnylam |
| Givlaari™ (Givosiran) | 2019 | Adult patients with acute hepatic porphyria | aminolevulinate synthase 1 (ALAS1) | RNA - GalNAc-conjugation | Alnylam |
| Oxlumo™ (Lumasiran) | 2020 | Adult and pediatric patients with primary hyperoxaluria type 1 | Hydroxyacid oxidase 1 (HAO1) | RNA - GalNAc-conjugation | Alnylam |

bind to ASGP receptors on the cell surface, resulting in rapid endocytosis mediated by clathrin-coated vesicles[71,72]. Furthermore, GalNAc is described as a powerful, long-lasting, and low toxicity approach for the delivery of siRNAs and ASOs and, compared to LNPs (like ONPATTRO®), GalNAc-siRNA conjugates are more straightforward to synthesize and refine[73]. The approval of AMVUTTRA™ is based on the encouraging results of the HELIOS-A phase 3 clinical study (NCT03759379) that enrolled 164 patients (vutrisiran, $n = 122$ vs patisiran reference group, $n = 42$); the group of 77 patients treated with the placebo in the APOLLO trial was used as an external control. The results showed that vutrisiran did elicit clinically relevant improvements of the signs and symptoms of polyneuropathy in half of the patients, in comparison with the external placebo group[74,75]. The benefits of vutrisiran treatment were generally similar to those observed with patisiran. However, vutrisiran is administered subcutaneously every 3 months, whereas patisiran is administered by intravenous infusion every 3 weeks and requires premedication prior to reduce the risk of infusion reactions. Based on the presumption of benefit of the route of administration of vutrisiran for the patients, AMVUTTRA™ was approved for the treatment of patients with hATTR if available treatments cannot be administered. Two phase III trials are ongoing on patisiran (APOLLO-B NCT03997383)[76] and vutrisiran (HELIOS-B NCT04153149)[77,78] for the treatment of hATTR amyloidosis with cardiomyopathy.

Based on the enhanced stability, delivery and duration of responses of GalNAc-siRNA conjugate platforms, Alnylam Pharmaceuticals developed two other RNAi therapeutics that were recently approved for the treatment of acute hepatic porphyria (AHP) and primary hyperoxaluria type 1 (PH1). AHP is an inherited metabolic disorder caused by variants in delta-aminolevulinic acid dehydratase (ALAD) gene. ALAD mutations drive to the accumulation of toxic hepatic heme intermediates. The GalNAc-siRNA GIVLAARI® (Givosiran) was designed to inhibit delta aminolevulinic acid synthase 1 (ALAS1) expression, which reduces the production of these toxic intermediates[79]. In the phase 3 ENVISION clinical trial (NCT03338816), that enrolled a total of 94 patients (48 in the givosiran group and 46 in the placebo group), subcutaneous injections of GIVLAARI® resulted in a significant reduction of porphyria attack rate per year (3.2 in the givosiran-treated group vs 12.5 in the placebo group). In addition, sustained reductions in levels of delta-aminolevulinic acid and beneficial effects across a broad range of acute and chronic disease manifestations were observed[80]. Based on these observations GIVLAARI® was approved by the FDA and EMA in 2019.

PH1 is a rare inherited disorder characterized by the overproduction of oxalate[81], which forms insoluble calcium oxalate crystals that accumulate in the kidney and other organs such as heart, eyes, bones, and skin. Individuals with PH1 are thus at risk for recurrent nephrolithiasis, nephrocalcinosis or end-stage renal disease[81]. The GalNAc-siRNA developed to treat PH1 (OXLUMO®, Lumasiran) reduces hepatic oxalate production by targeting glycolate oxidase (GO)[82]. Decreased GO enzyme levels in the body result in reduction of plasma and urinary oxalate levels, the underlying cause for the symptoms in patients with PH1. A phase 3 trial (ILLUMINATE-A, NCT03681184) was conducted in a cohort of 39 patients (lumasiran group $n = 26$ and placebo group $n = 13$) with PH1 and relatively preserved kidney function who were 6 years of age or older. This study reported that

subcutaneous injections of OXLUMO® led to substantial reductions in urinary and plasma levels of oxalate with urinary oxalate levels reaching the normal or near-normal range in 84% of patients by the sixth month of the trial[83,84]. OXLUMO® was approved by both FDA and EMA in 2020 for the treatment of PH1[85]. Recently, the results of two phase 3 studies evaluating OXLUMO® efficacy in a cohort of children under 6 years old (ILLUMINATE-B, NCT03905694, $n = 18$)[86] and patients with PH1 associated with advanced kidney disease or kidney failure (ILLUMINATE-C, NCT04152200, $n = 21$)[87] have been published. The results of ILLUMINATE-B demonstrated that, at month 6, OXLUMO® significantly reduces urinary oxalate levels in young children (9/18 patients achieved normal range levels) and the mean percent reduction in plasma oxalate levels was 31.7%. In patients with advanced disease (ILLUMINATE-C), the mean reductions in plasma oxalate levels were 33,3% for the patients that haven't received hemodialysis at enrollment and 42.4% for those who received hemodialysis, with acceptable safety profile at 6 month[86,87].

Alternative siRNA-based therapeutics to treat primary hyperoxaluria (PH) disorders (including the three subtypes PH1, PH2 and PH3) have been developed. The most advanced candidate is nedosiran (Dicerna® pharmaceuticals) a siRNA designed to prevent the production of the hepatic lactate dehydrogenase (LDH) enzyme, involved in the glyoxylate-to-oxalate conversion[88]. Encouraging clinical phase 1 and 2 studies results have been reported (PHYOX-1 NCT03392896, PHYOX-2 NCT03847909 and PHYOX-4 NCT04555486 trials) evaluating its safety profiles and efficacy on patients with PH1, PH2 and PH3[89–91]. Results from the phase 3 study (PHYOX-3 NCT04042402) are still awaited[92] and further phase 2 and 3 studies in PH2 and PH3 patients, pediatric patients, and patients with advanced kidney disease are needed to get the FDA approval[93].

*Antisense oligonucleotides (ASOs)*: ASOs are short, synthetic, single-stranded oligodeoxynucleotides that can alter RNA and reduce, restore, or modify protein translation. ASO-based therapeutics functionalize through plethora of distinct mechanisms of action belonging to two major categories: RNase H-dependent and RNase H- independent (steric block) oligonucleotides[94]. The endogenous RNase H enzyme, active in both cytoplasm and nucleus of the cells, recognizes and catalyzes the cleavage of RNA-DNA heteroduplexes formed when DNA-based oligonucleotides bind to their complementary transcripts[95] (Fig. 3). The RNase H-dependent ASOs trigger the degradation of the target mRNA, thereby suppressing its translation. This therapeutic approach is very useful to silence disease-causing genes[96]. Steric block oligonucleotides do not modulate gene expression *via* the action of the RNase H enzyme, as they do not form RNase H substrates. Steric block ASOs are typically RNA-based oligonucleotides that directly bind to their target pre-mRNA or mature mRNA and sterically deny other molecules access. They can be employed for various applications such as the modulation of alternative splicing[97], mRNA translation inhibition[98] or promotion[99] and shifting of polyadenylation signals to increase transcript stability[100]. Steric block ASOs are mainly used for their abilities to alter splicing decisions. These Splice-Switching antisense Oligonucleotides, or SSOs, are developed to specifically hide a splice recognition site in order to keep (exon inclusion) or exclude (exon skipping) specific exons (Fig. 3 panel 3). This therapeutic approach, also called splicing

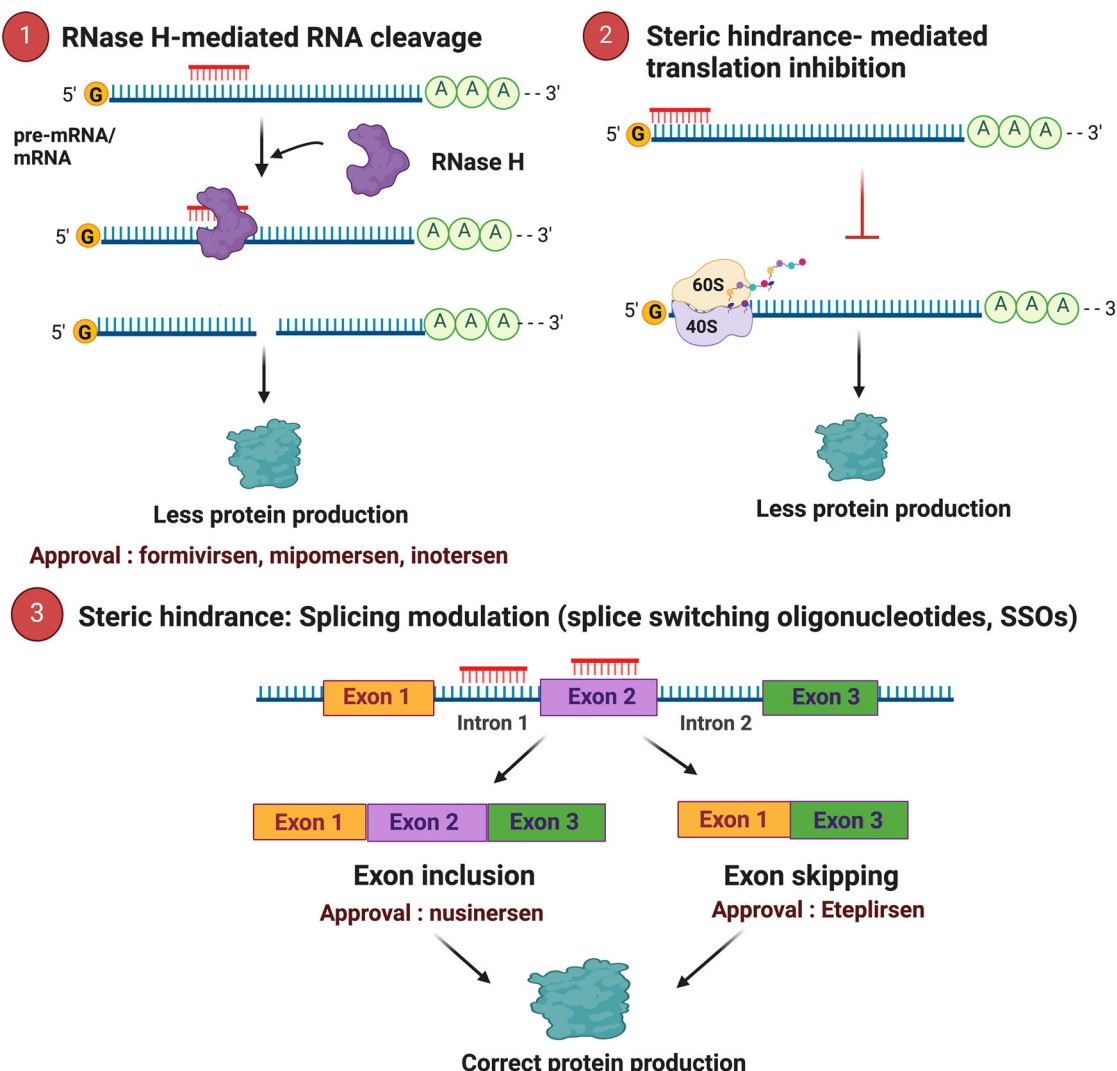

**Fig. 3 | Main mechanisms of Antisense oligonucleotide (ASO).** (1) In RNase H-mediated RNA cleavage, heteroduplex formation between an ASO and an mRNA or pre-mRNA activates RNase-H, leading to mRNA/pre-mRNA degradation. (2) Steric hindrance based ASOs binding to mRNAs can disrupt the interaction of mRNAs with the 40S ribosomal subunit or interfere with their assembly on the 40S or 60S ribosomal subunits, resulting in translational arrest. (3) Splice-switching oligonucleotides or SSOs can modify gene expression by modulating pre-mRNA alternative splicing, resulting in exon inclusion or exon skipping. Created with BioRender.com (Agreement number : CO26ML0CXF).

corruption, is very useful in reducing the levels of disease-causing protein isoforms and promoting the production of the beneficial ones.

Molecules destined for clinical applications are heavily modified to enhance nuclease resistance, reduce toxicity and improve RNA target binding affinity. Such modifications involve the replacement of one of the non-bridging oxygen atoms by other atom or chemical group such as methyl one. Phosphorothioate oligonucleotides with the oxygen substituted by sulfur atom are the most investigated ASOs. The second generation ASOs include modification of the hydroxyl group at 2′ position of ribose which is replaced with a fluorine atom or methyl and methoxyethyl groups (ME and MOE). Third generation ASOs such as phosphoramidates, morpholino phosphoroamidates (PMOs) as well as locked nucleic acid (LNA) contain modification in phosphate groups. Lastly, nucleobase modifications can also be incorporated into ASO but only replacing cytosine with 5-methylcytosine has proved benefits.

The first ASO approved for treating humans (VITRAVENE®, fomivirsen) was developed in 1998 by Isis pharmaceuticals to treat cytomegalovirus (CMV) retinitis in HIV-infected patients who cannot tolerate or have not responded to other drugs[101–104]. This 21-bases synthetic phosphorothioate oligonucleotide, administered through intravitreal injections, is complementary to the transcript of the major immediate early 2 (IE2), which is necessary for CMV replication. The discovery of new highly active antiretroviral therapies, reducing the incidence of opportunistic infections in individuals with HIV in the early 2000s, led to the withdrawn of the drug by the FDA in 2001 and it was taken off the market by Novartis in 2002. Nevertheless, the success of VITRAVENE® provided proof-of-concept of the clinical promise of treatments based on ASOs and since then, the list of therapeutic approaches based on the use of ASOs to treat genetic diseases has grown continuously. Eight antisense oligonucleotide-based drugs have been approved for marketing in February 2022: Kynamro®, SPINRAZA®, TEGSEDI®, WAYLIVRA®, Exondys51®, Vyondys53®, Amondys45®and Viltepso®, the last four agents have been approved for the treatment of Duchenne Muscular Dystrophy (DMD) (Table 3).

DMD is a fatal X chromosome-linked inherited recessive disorder, caused by the loss or reduced levels of functional dystrophin, a key muscular protein encoded by the *DMD* gene. This disorder is characterized by infantile-onset progressive muscle weakness leading to the death of the patients in their 20 s. Most DMD patients (~80%) carry large (affecting one or more exons) deletions and duplications. The remaining 20% of DMD patients are affected by nonsense mutations, small insertions or deletions

**Table 3 | Approved ASO-based therapies for genetic diseases**

| Trade name (drug name) | Strategy | Approval year | Disease | Vector - modification | Targeted sequence | Company |
|---|---|---|---|---|---|---|
| Vitravene™ (Fomivirsen) | ASO | 1998†2002 | Cytomegalovirus retinitis | RNA - Phosphorothioate | CMV IE2 | Ionis Pharmaceuticals |
| Kynamro™ (Mipomersen) | ASO | 2013†2018 | Homozygous familial hypercholesterolemia | RNA - Phosphorothioate/2'O-methoxyethyl | ApoB-100 | Sanofi and Ionis Pharmaceuticals |
| Exondys51™ (Eteplirsen) | ASO | 2016 | Duchenne Muscular Dystrophy | RNA - Phosphorodiamidate morpholino | Exon 51 of DMD | Sarepta Therapeutics |
| Spinraza™ (Nusinersen) | ASO | 2016 | Spinal Muscular Atrophy | RNA - Phosphorothioate/2'O-methoxyethylated | Exon 7 of SMN2 | Biogen |
| Tegsedi™ (Inotersen) | ASO | 2018 | Hereditary Transthyretin-related Amyloidosis | RNA - Phosphorothioate/2'O-methoxyethylated | Transthyretin (TTR) | Ionis Pharmaceuticals |
| Vyondys53™ (Golodirsen) | ASO | 2019 | Duchenne Muscular Dystrophy | RNA - Phosphorodiamidate morpholino | Exon 53 of DMD | Sarepta Therapeutics |
| Waylivra™ (Volanesorsen) | ASO | 2019 | Adult Familial Chylomicronemia syndrome | RNA - Phosphorothioate/2'O-methoxyethyl | ApoC3 | Ionis Pharmaceuticals |
| Viltepso™ (viltolarsen) | SSO | 2020 | Duchenne Muscular Dystrophy | RNA - Phosphorodiamidate morpholino | Exon 53 of DMD | Nippon Shinyaku |
| Amondys45™ (casimersen/SRP-4045) | ASO | 2021 | Duchenne Muscular Dystrophy | RNA -Phosphorodiamidate morpholino | Exon 45 of DMD | Sarepta Therapeutics |
| Milasen™ | ASO | 2019 | Batten Disease | RNA -Phosphorothioate/2'-O-methoxyethyl | i6.SA cryptic splice-acceptor site in MFSD8 gene | Boston Children Hospital |
| Wainua™ (eplontersen) | ASO | 2023 | Transthyretin-mediated amyloidosis | GalNAC-conjugated antisense oligonucleotide | Transthyretin (TTR) | Ionis Pharmaceuticals + AstraZeneca |
| Qalsody™ (tofersen) | ASO | 2023 | SOD1 mutations associated amyotrophic lateral sclerosis (ALS) | RNA - Phosphorothioate/2'O-methoxyethylated | Superoxide dismutase 1 (SOD1) | Ionis Pharmaceuticals + Biogen |

†Withdrawal of approval.

and splice site mutations of the *DMD* gene, causing the insertion of a premature stop codon. The rationale underlying the use of oligonucleotides-based therapeutic approaches for DMD, came partly from the observation of the mutations found in Becker muscular dystrophy patients (BMD). Indeed, BMD is characterized by in-frame deletions which allow the production of a partially truncated, but functional, version of dystrophin resulting in a milder form of muscular dystrophy[105]. These observations led to the development of exon skipping therapies to treat DMD aiming to force the cells' machinery to skip one or more exon(s) in the *DMD* gene's pre-mRNA. This therapeutic approach promotes the production of a smaller, but semi-functional, dystrophin protein. The most frequently deleted exons in DMD patients are exons 51, 53 and 45 mutated in approximately 20%, 13% and 12% of patients, respectively[106,107]. SSOs targeting these top three exons have been mainly developed, however, other exons would need to be targeted by specific ASOs to treat each subset of DMD patients.

Exondys51® (Eteplirsen, developed by Sarepta Therapeutics) is a third generation phosphomorpholidate morpholino (PMO) SSO designed to enable the skipping of exon 51, thus, avoiding the production of the mutated dystrophin protein. Eteplirsen is applicable on 14% of the entire DMD-patient population which is, by far, the largest patient population that could be targeted by a single exon skipping based therapeutics. A phase 2 study (NCT01396239), that enrolled 12 DMD patients (8 patients treated with Eteplirsen and 4 patients in the placebo-treated cohort), demonstrated a significant increase in shorter but functional dystrophin protein levels in patients' muscle biopsy samples after 48 weeks[108]. Treatments with Eteplirsen delayed muscle deterioration, prolonged ambulation, and preserved pulmonary function of patients with DMD. Based on these encouraging results, Exondys51® was granted accelerated approval by the FDA in 2016, making it the first FDA-approved drug for DMD[109].

Since then, three other SSO-based therapies have been approved. Two were also developed by Sarepta Therapeutics: Vyondis53® (Golodirsen)[110] approved in 2019, for exon 53 skipping and Amondys45® (Casimersen) approved in 2021[111], for exon 45 skipping. Viltepso® (viltolarsen), enabling exon 53 skipping[112], was developed by the Japanese laboratory Shinyaku and the National Center of Neurology and Psychiatry (Kodaira, Japan), it was approved in the United States in August 2020[113].

SSO-based therapeutics has also been developed to treat SMA with the development of SPINRAZA® (Nusinersen marketed by Biogen). Indeed, in patients with SMA the *SMN2* gene (a *SMN1* gene homolog) differs by a C-to-T variant in the exon 7 resulting in the exclusion of this exon in SMN2 mRNA. The alternative splicing process of *SMN2*, leads to the production of a smaller and less efficient SMN protein. Nusinersen is an 18 mer 2'-O-MOE SSO that promotes the inclusion of exon 7 in SMN2 mRNA by targeting the Intronic Splicing Silencer N1 (ISS-N1) in intron 7. Preclinical studies in cell cultures and transgenic mice have first proven that modulation of SMN2 exon 7 splicing using SSOs were effective to increase SMN protein levels[114]. Further clinical studies have subsequently showed remarkable improvements in motor functions and patient survival leading to nusinersen approval by EMA and FDA in 2016[115–118]. However, it is now well-established that its efficiency varies among patients and recent studies suggested that treatment initiation at early stages of the disease provides better responses[119–121].

ASO-based therapies for metabolic disorders have also been extensively investigated and three drugs have been successfully developed by the Ionis Pharmaceuticals company. These drugs reduce targeted RNA *via* RNase H1-dependent degradation of the RNA/DNA hybrid. Kynamro® (Mipomersen, Ionis Pharmaceuticals), a 2'-O-MOE phosphorothioate ASO, was approved by the FDA in 2013 for the treatment of homozygous familial hypercholesterolemia, a rare life-threatening condition characterized by markedly elevated circulating levels of low-density lipoprotein cholesterol (LDL-C) and accelerated, premature atherosclerotic cardiovascular disease[122,123]. This disorder is mainly caused by mutations in three genes encoding the LDL receptor, apolipoprotein B (APOB), or proprotein convertase subtilisin/kexin type 9 (PCSK9)[124]. Mipomersen targets the *APOB* mRNA, an essential component of LDL particles involved in

atherosclerotic disease progression, and is administered *via* subcutaneous injection[125]. Reduced levels of apolipoprotein B (apo B) result in decreased production of LDL-C, and total cholesterol[126]. However, due to limited clinical effects and safety concerns related to its hepatotoxicity, Mipomersen was discontinued in 2018.

Waylivra® (Volanesorsen, Formerly IONIS-APOCIIIRx) is an antisense drug, used for the treatment of familial chylomicronemia syndrome (FCS), a genetic disorder characterized by high levels of triglycerides and a greatly elevated prevalence of acute pancreatitis[127]. FCS is caused by impaired function of the lipoprotein lipase leading to impaired clearance of triglyceride-rich lipoproteins from plasma and to the accumulation of triglycerides in the blood[128]. Volanesorsen is the first drug that targets *APOC3* mRNA, a major determinant of plasma triglyceride concentration, that regulates both the lipoprotein lipase-dependent and -independent pathways responsible for the clearance of triglyceride-rich lipoproteins[129]. The phase 3 APPROACH trial demonstrated that subcutaneous treatment with volanesorsen resulted in a significant reduction in mean plasma triglyceride levels, in patients with FCS[130,131]. Despite the encouraging results of the APPROACH trial (NCT02211209), the high incidence of thrombocytopenia restricts its clinical application. Other ASO-based drugs targeting APOC3 are currently in early stages of research, such as Olezarsen which is a stronger APOC3 antagonist. In the phase 1/2a trail AKCEA-APOCII-LRx (NCT02900027), olezarsen treatments led to a significant reduction of APOC3 and triglycerides levels and, most importantly, no case of thrombocytopenia was reported[132].

TEGSEDI® (Inotersen, formerly IONIS-TTRRx/ISIS 420915), is a 2′-O-MOE–modified ASO that prevents the hepatic production of transthyretin protein. In a phase 3 trial (NCT01737398) reported that a once-weekly subcutaneous injection of inotersen improved the course of neurologic disease and quality of life in patients with stages I- and II-related hATTR polyneuropathy. Inotersen was approved by the FDA and the EMA in 2018, however, some safety concerns were noted and this treatment is now contraindicated for patients with thrombocytopenia ($<100,000/\text{m}^3$) and glomerulonephritis[133,134]. Several NA-based drugs such as siRNAs (Patisiran and Vutrisiran) and ASO (inotersen) were approved by the FDA for the treatment of patients with hATTR amyloidosis. However, to develop additional treatment options for these patients with better clinical benefits, Astrazeneca and Ionis Pharmaceuticals developed Eplontersen a GalNAC conjugated ASO that targets and degrades the *TTR* mRNA in the liver of patients with hATTR polyneuropathy[135,136]. The NEURO-TTRansform phase 3 trial (NCT04136184) was initiated in December 2019 on 168 patients with hATTR polyneuropathy across 16 countries. The eplontersen treated group was compared to the historical inotersen placebo group. Among the 168 included in the clinical assay, 144 were treated with eplontersen and 24 were randomized to the inotersen control group (60 placebo patients total). The results showed that an 81.7% reduction of TTR serum levels in the eplontersen treated group compared to 11.2% in the placebo group. Major adverse events occurred in 6% of the patients treated with eplontersen at week 66[135]. Based on these results, eplontersen (WAINUA™) was approved by the FDA in December 2023[137].

QALSODY™ (tofersen), a novel ASO-based therapy, received an accelerated approval from the FDA in April 2023, however the EMA and Medicines and Healthcare products Regulatory Agency (MHRA) approvals are still pending[138]. Tofersen was developed to target the mutant superoxide dismutase 1 (SOD1) protein, responsible for 10–20% of hereditary amyotrophic lateral sclerosis (ALS)[139,140]. ALS is a progressive and fatal neurodegenerative disorder that affects motor neurons leading to muscle paralysis. The phase 3 VALOR trial (NCT02623699) enrolled 108 patients diagnosed with ALS and confirmed mutations in the *SOD1* gene to investigate the clinical benefits of tofersen treatment (randomization ratio of 2:1, 72 patients received the treatment, and 36 patients were assigned in the placebo group)[141]. Tofersen was directly injected in the cerebrospinal fluid (CSF) of the patients. After 28 weeks of treatment, despite great reductions of SOD1 protein levels in CSF, the improvements of the clinical symptoms in the tofersen treated cohort were not statistically significant compared to

the placebo cohort[141]. The results of the VALOR open-label extension (95 participants) suggested that earlier intervention may be necessary to observe notable clinical benefits of the tofersen therapy and overall quality of life improvements in patients with SOD1-ALS. Tofersen treatment benefits, in presymptomatic or at-risks individuals, are now being investigated in the ATLAS trial (NCT04856982)[142].

Hundreds of ASO-based therapies are being investigated and some attracted great attention but still too few have been approved. This is for instance the case of the phase 1/2a clinical trial launched by Roche to evaluate the efficiency of repeated intrathecal administration of an ASO-targeting huntingtin protein (HTT) (Tominersen; previously IONIS-HTTRx or RG6042) to treat Huntington's disease. Tominersen, was designed to reduce the production of all forms of the huntingtin protein (HTT), including its mutated variant, mHTT. While it was clearly demonstrated in a phase 1/2a study that this drug successfully reduced cerebrospinal fluid levels of mutant HTT protein[143], the phase 3 trial GENERATION HD1 (NCT03761849) was halted because it failed to show higher clinical efficacy than placebo[144,145]. Yet, exploratory post-hoc analyses suggest that Tominersen treatment initiation in younger adult patients with lower disease burden may provide better effects. Roche is currently in the early stages of designing the phase 2 clinical trial (NCT05686551) to explore different doses of the ASO in this subpopulation of patients[146].

### CRISPR/Cas9 gene editing system (clustered regulatory interspersed short palindromic repeats)

The clustered regularly interspaced short-palindromic repeat (CRISPR)-associated protein 9 (CRISPR/Cas9) nuclease system provide an innovative technology for genome editing. This emerging technique allows scientists to change the DNA structures and modify gene functions[147]. The CRISPR system was first reported by Ishino and colleagues in 1987 as unusual repeats of DNA sequences in *Escherichia coli* (*E. coli*)[148]. Further studies described the CRISPR system as an ancient bacterial-based immune system used to prevent viral infection by specifically recognizing and destroying viral DNA or RNA. Through a specific sequence-recognition of the single guide RNA (sgRNA) with the target sequence of genome and the presence of a protospacer-adjacent motif [PAM] sequence, CRISPR associated nucleases (Cas proteins) are recruited for creating target-specific double-strand breaks (DSBs)[149] (Fig. 4). The CRISPR/Cas9 system is the most widely used genome editor. Cas9 nucleases are able to efficiently target a wide range of DNA sequences directed by a sgRNA to create DSBs that are mainly repaired by endogenous repair pathways including the nonhomologous end joining (NHEJ) and the homology-directed repair (HDR). CRISPR/Cas9 technology has raised great excitement as a potential one-time therapy for genetic diseases (Table 4 and Supplementary Table 1) and its mechanistic resolution has been awarded by the Nobel Prize in Chemistry in 2020.

The first in human clinical trial involving CRISPR/Cas9 system was initiated in 2016 to treat advanced non-small cell lung cancer and aimed to knockout the immune checkpoint Programmed cell Death-1 (PD-1) in autologous T-cells (NCT02793856)[150,151]. However, the first human assay evaluating CRISPR/Cas9 editing technology to treat a genetic disorder was initiated in 2018 for patients with β-thalassemia and SCD (NCT03655678)[152]. Several studies reported that elevated levels of fetal hemoglobin, was associated with reduced mortality in patients suffering from these blood disorders[153,154]. The CRISPR-based ex vivo cell therapy exagamglogene autotemcel (exa-cel, formerly called CTX-001) has been developed to reactivate production of fetal hemoglobin in CD34+ hematopoietic stem and progenitor cells (HSPCs) (Vertex Pharmaceuticals Incorporated and CRISPR Therapeutics AG)[155]. Early results from two Phase I/II studies, evaluating Casgevy™ (exa-cel) transfusion efficacy, demonstrate significant increases in total hemoglobin and fetal hemoglobin in patients with transfusion-dependent β-thalassemia (TDT) (CLIMB THAL-111, NCT03655678) and patients with SCD (CLIMB SCD-121, NCT03745287)[156]. Indeed, the results of the CLIMB SCD-121 trial, show that in 93.5% of the patients ($n = 31$) the severe vaso-occlusive events were resolved[157]. Moreover, increased levels of fetal hemoglobin occurred early

**Fig. 4 | The main principles of CRISPR/Cas9 genome editing.** CRISPR-associated nucleases (Cas proteins) are recruited to generate target-specific double-strand breaks (DSBs) through specific sequence recognition of the single guide RNA (sgRNA) with the target sequence of the genome and the presence of a protospacer-adjacent motif [PAM] sequence. DSBs are mainly repaired by endogenous repair pathways, including nonhomologous end joining (NHEJ) and homology-directed repair (HDR). NHEJ facilitates joining two DNA fragments without the need for exogenous homologous DNA, resulting in small random insertions or deletions at the cleavage site. Consequently, the expression of the target gene is disrupted due to the production of frameshift mutations or premature stop codons during NHEJ-based repair. On the other hand, HDR facilitates precise gene insertion or replacement through the presence of donor DNA that contains a desired sequence and sequence homology at the DSB site. Created with BioRender.com (Agreement number : MB26ML0B76).

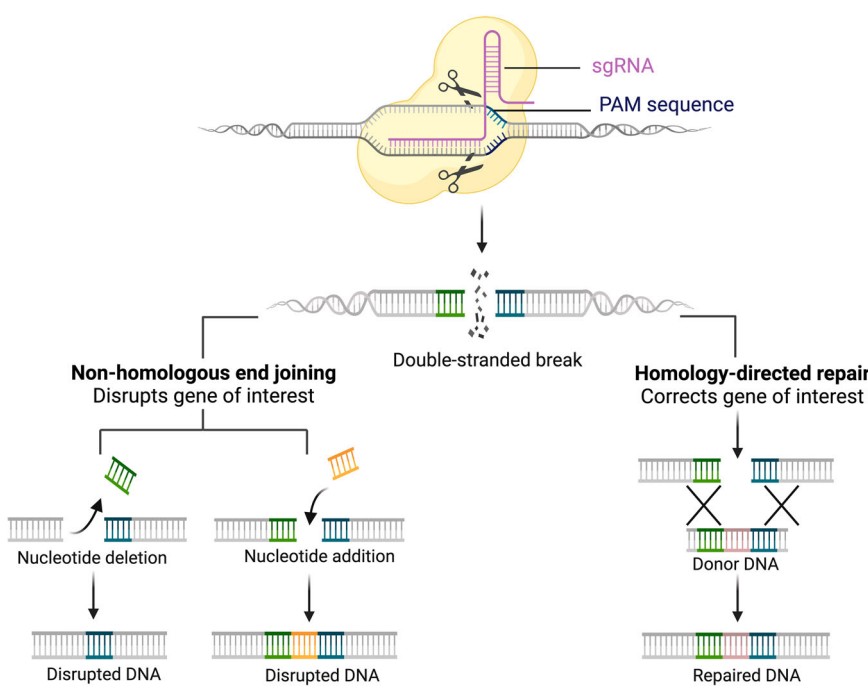

## Table 4 | Approved CRISPR-based medicine

| Trade name (drug name) | Approval year | Strategy | Targeted gene | Disease | Company |
|---|---|---|---|---|---|
| Casgevy™ (Exagamglogene autotemcel) | 2023 | Ex vivo-Autologous transfusion | *BCL11A* | β-thalassemia Sickle Cell Disease | Vertex Pharmaceutical |

and were maintained for at least 12 months during the 24-month follow-up period, and Casgevy™ safety profile was consistent with that of autologous HSPCs transplant[155,157]. In the CLIMB THAL-111 trial, 42 patients with TDT received a single dose of Casgevy™ and 39 were transfusion-free for at least one year[155]. In November 2023, the MHRA approved Casgevy™ for the treatment of patients with SCD and TDT ages 12 and older, who are eligible for a stem cell transplant but lack a donor, making the United Kingdom the first country to approve a CRISPR-based medicine[158] (Table 4). The green light from the MHRA represents a major breakthrough in the field. On December 8th 2023, the FDA announced the approval of the first CRISPR-based therapeutics, Casgevy™, for the treatment of SCD patients, ages 12 and older, who are experiencing recurrent vaso-occlusive crisis[48,159]. However, the FDA noted that patients receiving Casgevy™ will be followed in a long-term study to confirm its safety and efficiency. More recently, on December 15th 2023, the EMA approved Casgevy™ for the treatment of TDT and severe SCD in patients ages 12 and older for whom haematopoietic stem cell transplantation is appropriate and a suitable donor is not available[160].

The first in vivo CRISPR-Cas9 genome editing clinical trial was initiated in 2019 (NCT03872479) to evaluate the safety, tolerability and efficacy of AGN-151587 (EDIT-101 developed by Editas Medicine, Inc) in participants with Leber Congenital Amaurosis type 10 (LCA10)[161,162]. LCA10 is an eye disorder, affecting the patient's retina, caused by a mutation in the *CEP290* gene leading to severe visual impairment. EDIT-101 is an AAV-mediated CRISPR-Cas9 system aiming the restoration of CEP290 protein production through subretinal injection in patients with LCA10. The first results from this study are expected in 3 years (Supplementary Table 1).

Many other clinical trials, aiming to investigate in vivo CRISPR/Cas9 genome editing, have been initiated and are currently recruiting participants. An LNP-mediated CRISPR/Cas9 system (NTLA-2001 developed by Intellia therapeutics) has been developed to reduce TTR protein levels in the treatment of hATTR amyloidosis with polyneuropathy or transthyretin amyloidosis-related cardiomyopathy (NCT04601051)[163,164]. Interim data have been press released by Intellia therapeutics recently. They reported that

NTLA-2001 induced sustained reduction of mean serum TTR levels in the cardiomyopathy arm of the study and was well-tolerated at both tested doses[165]. Intellia therapeutics also initiated a phase I/II clinical study to evaluate its second in vivo genome editing candidate (NTLA-2002), for the treatment of hereditary angioedema (HAE) (NCT05120830)[166,167]. NTLA-2002 is an LNP-mediated CRISPR-Cas9 system designed to inactivate the *kallikrein B1* (*KLKB1*) gene, which encodes for the kallikrein precursor protein (prekallikrein)[167]. Updated data, presented at the American College of Allergy, Asthma & Immunology (ACAAI) 2022 Annual Scientific Meeting, showed that patients who received a single dose of NTLA-2002 were swelling attack-free for up to 10 months and that the treatment was well-tolerated. In September 2022, the FDA has granted orphan drug designation for NTLA-2002, for the treatment of HAE[168]. VT-1001 is a phase Ib trial that evaluates the safety and pharmacodynamic profile of VERVE-101, an LNP-mediated CRISPR system developed by Verve Therapeutics, when administered to patients with heterozygous familial hypercholesterolemia (HeFH), atherosclerotic cardiovascular disease (ASCVD), and uncontrolled hypercholesterolemia (NCT05398029)[169]. VERVE-101 is designed to prevent the expression of the *PCSK9* gene in the liver of patients with familial hypercholesterolemia and cardiovascular disease (Supplementary Table 1).

The recent success of Casgevy™ (Vertex) is paving the way for the utilization of gene editing therapies in clinic. More and more CRISPR-based therapeutics will be clinically investigated in the next few years. However, due to the technology novelty, long-term adverse effects merits careful examination.

## The key issue of delivery strategies

One of the great challenges to extend the clinical use of NA-based drugs includes the development of carriers that would effectively deliver them to the target tissues, cells, and organelles[170]. Therapeutics based on nucleic acids are susceptible to enzymatic degradation during their journey in the systemic circulation[171,172]. Indeed, regardless of the route of administration, naked nucleic acids are rapidly cleared from the organism due to the

immune system, and their degradation by endogenous nucleases. In addition, nucleic acids are negatively charged and hydrophilic which impede their passive diffusion across the plasma membranes. As a result, nucleic acids require delivery systems development to protect them from degradation and to deliver them to the target cells for efficient uptake[78]. The different delivery systems for NA-based treatments are well documented in the literature, including viral- and non-viral-based delivery techniques[18,171–174].

Viral-based vectors received huge attention and 80% of gene therapies that are approved and in clinical trials are using retroviruses (including lentiviruses), Adeno-Associated Viruses (AAVs), Adenoviruses (Ads), or Herpes Simplex Viruses (HSVs) to deliver the genetic material[175–177] (Supplementary Fig. 1). The selection of the vector depends on several factors such as its ability to enter the target cells efficiently, its capacity to transfer the genetic material into the nucleus, the size of the genetic material to be delivered, its safety (level of pathogenicity) and immunogenicity[178,179].

Ads and AAVs are typically used for in vivo delivery consisting in direct administration of the drugs to the patients, the genetic material will remain in episome, this delivery strategy is mainly used for vaccines against infections and cancer treatments[180,181]. To treat patients with RD, AAV-based vectors are of the most commonly used viral vectors to deliver gene replacement therapeutics because of their long-term and effective expression of the transgenes, their low immunogenicity and their excellent safety profile in both pre-clinical and clinical investigations[182–184]. More than 200 AAV-based NA therapeutics are currently being investigated in clinical trials, and as previously mentioned, 3 AAV-based gene replacement therapies reached the commercial market: Glybera, Luxturna, and Zolgensma[36,43,185] (Table 1).

Retroviruses are more applicable for ex vivo gene therapy approaches where the foreign genetic material is delivered into hematopoietic cells or other types of stem cells and integrated into the host genome which is advantageous for long-term transgene expression[175,184]. However, the random integration of the transgenes when using retrovirus-based vectors is a major concern and leukemia development has been reported in clinical assays on X-linked SCID patients[184]. In the clinical trial investigating Strimvelis efficacy in the treatment of ADA-SCID patients, on 50 tested patients only 1 case of leukemia has been reported leading to its approval in 2016[186]. LVs, which belong to the retroviruses family, can incorporate genetic material into the host chromosomes in semi-random manner and display low toxicity[183]. Unlike simple retroviruses, LV vectors could enter in both dividing and non-dividing cells. Due their improved biosafety, a numerous of clinical trials based on LVs-based vectors has been implemented for the treatments of various diseases including genetic disease such as β-thalassemia[187], cerebral adrenoleukodystrophy[188], metachromatic leukody[189]. The gene therapy Zynteglo, using LV-based vectors for the transgene delivery, was approved in 2019. The recently approved Lyfgenia™ (lovo-cel), also works by using a LV vector to increase the levels of normal hemoglobin in patients with SCD (Table 1)[190].

The extensive understanding of Herpes simplex virus (HSV) sequences and advancements in molecular techniques have paved the way for utilizing HSV as a versatile tool in various applications related to human health. Herpes simplex virus 1 (HSV-1) possesses several characteristics that make it an attractive candidate for therapeutic gene delivery including: episomal delivery, a broad tissue tropism, high transduction efficiency, large transgene capacity and the ability to resist immune clearance via the inhibition of innate and adaptive anti-viral immunity[177]. Attenuated or defective HSV1-based vectors have thus been developed and shown promising results in treating various peripheral and central nervous system diseases[191]. Yet, this approach has so far only been validated clinically, in 2023, in dermatology for the treatment of rare blistering skin disorder dystrophic epidermolysis bullosa[28,32]. Vyjuvek™ (beremagene geperpavec) is the first FDA approved HSV1-based gene therapy[30]. The advantages and disadvantages of each of the mainly used viral vectors for NA therapeutics delivery, are summarized in the Supplementary Table 2.

Non-viral vectors are less popular due to their lower delivery efficacy and gene expression duration. However, viral vectors raise some safety concerns related to their viral origin, oncogenic potentiate, and immunogenicity[18,192,193]. Most of the RNA therapeutics, contributing for a majority of FDA or EMA approved NA-based therapeutics, are not conjugated with any delivery platforms, as enormous effort has been made in identifying chemical modifications which improve the stability of nucleic acids and prevent their degradation[172,173,194]. The structure of the nucleic acids can be modified at different levels: the nucleobase, the carbohydrate, and the phosphodiester linkage. The most employed modifications in approved ASOs include the phosphorothioate (PS) modification in the phosphodiester linkage and the 2′-O-methoxyethyl (2′-O-MOE) in the sugar group. These modifications, that significantly increase the stability, binding affinity, and reduce the immunogenicity of ASOs, were utilized for the development of Kynamro™[126], Spinraza™[116], Tegsedi™[134], and Waylivra™[131] for the treatment of various genetic diseases. The approved ASO-based treatments Exondys51™[108], Vyondys53™[110], Viltepso™[64] and Amondys45™[111] contain the backbone chemical modification phosphorothioamidate morpholino oligomer (PMO) which improve drug efficacy and specificity, nuclease resistance, solubility in water, and reduce production costs[173] (Table 3).

Beside these chemical modifications of the internal structures of nucleic acids, molecules of various nature (lipids, sugars…) can be conjugated with NA therapeutics to effectively enhance their stability and cell penetration, and also facilitates their accumulation in specific organs. As mentioned above, three siRNAs and one ASO linked to GalNAc moiety received green lights of FDA and/or EMA: Givlaari™[79], Oxlumo™[14], Amvuttra™[69] and Wainua™ (Tables 2 and 3). GalNAc conjugates are clinically convenient as they can be self-administered subcutaneously, resulting in rapid absorption, high uptake, and long half-life.

With the development of nanotechnologies, novel nanocarriers are gradually replacing the viral vectors. The commonly found nano-vehicles for NA therapies transportation are summarized in Supplementary Fig. 1 and are based on lipids (liposomes, solid lipid nanoparticles…), polymers (micelles, dendrimers, nanogels…) and inorganic materials including metal and non-metal nanoparticles[170,171,173,192]. The lipid-based nanoparticles (LNPs) utilization was reported to be effective in delivering drugs to the brain due to their ability to cross the blood-brain barrier (BBB)[195]. In particular, LNPs have been intensively studied for small molecules and NA-based drugs delivery[196] and recently attracted great attentions due to big success of mRNA-based vaccines against COVID-19[194,197]. Approved NA therapeutics using LNP vehicles include gene silencing based-siRNA Onpattro™[66], and mRNA vaccines against COVID-19 (Comirnaty/tozinameran and mRNA-1273)[198].

## The rate-limiting endosomal escape problem

The clinical successes of NA-therapeutics have raised a great interest and investment by pharmaceutical companies and biotechs. However, the major impediment that must be addressed before using NA-based therapies to treat widespread human diseases is the rate-limiting delivery issue of endosomal escape. As previously mentioned, the negative charges and hydrophilicity of NA prevents their passive diffusion across the cellular membranes and NA-based treatments are internalized into cells by endocytosis. Endosomes are intracellular vesicles delineated by a lipid bilayer barrier which entrap 99% of NA-therapeutics and allow 1% or less of these agents to enter the cytoplasm[199,200]. Although this very low rate has proven its efficacy in certain genetic disorders, enhancing endosomal escape would be required for the widespread use of NA-based therapeutics to treat human diseases. For the last decade, several groups tried to improve RNA therapeutics pharmacology by enhancing its transfer to the cytosol. One of the earliest approaches to enhance the release of RNA therapeutics from endosomal vesicles, was the use of endolytic small molecule agents such as chloroquine[201–203]. It was well known that the proton sponge ability of chloroquine and related agents can cause the rupture of the endosomal barrier through osmotic swelling[204]. Unfortunately, this class of agents

causes non-specific disruption of endosomal lipid bilayer and are efficient at high concentrations, resulting in clinically unacceptable toxicity[115,116]. More recently, many investigations focused on screening chemical libraries to identify small molecules that enhance RNA therapeutics release from endosomes[205–210]. To date, several compounds were reported to either increase cellular uptake or endosomal escape, however, they only displayed modest effects. The conjugation of cationic peptide domains with RNA therapeutics raised great hope in enhancing endosomal escape recently. Sarepta Therapeutics published results from a preclinical study investigating the efficiency of RC-1001, a Dmd exon 23-skipping PMO conjugated with a cationic peptide (PPMO), for the treatment of DMD. The results showed the PPMO treatment was more effective than a 10-fold greater dosing of its PMO counterpart in DMD mouse model[211]. Despite these encouraging pre-clinical results, the FDA placed, in June 2022, a clinical hold on the phase 2 clinical assay MOMENTUM (NCT04004065) evaluating an exon 51-skipping PPMO, SRP-5051 (Vesleteplirsen) on patients with DMD, after a patient experienced a serious adverse event with grade 3 hypomagnesemia, grade 4 potassium deficiency, muscular cramps, and mild-to-moderate tingling of the extremities. The FDA requested information on all cases of hypomagnesemia, including a small number of nonserious grade 2 cases. The clinical hold was lifted in September 2022 after information was provided by the Company to assess the adequacy of the risk mitigation and safety monitoring plan. Participant enrollment then resumed; the enrollment phase was completed in December 2022.

Moreover, the strategy based on the addition of cationic peptides or derivatives to RNA therapeutics is restricted to neutral backbone PMOs, as their combinations with the negatively charged siRNAs and ASOs lead to the formation of ionic aggregates. Another approach to solve this problem arises from the observation that viruses developed the ability to escape the endosomal compartment during cell infection[212]. One of the most studied virus is the lnfluenza virus and its surface hemagglutinin (HA) proteins that can disrupt the endosomal membrane locally resulting in the entrance of the viral capsid in the cytoplasm in a non-toxic manner[213]. It has been reported that the conjugation of the HA2 subunit of Influenza, used as an endosomal escape motif, with a siRNA therapeutics significantly increased its effectiveness in cancer cells[214]. However, proteins that belong to the enveloped capsules of viruses, like HA, are highly immunogenic and it may be challenging to move forward to clinical trials. An interesting strategy to improve the release of oligonucleotides in the cytoplasm, would be to directly manipulate the components of the endosomal machinery. Several studies focused on identifying the proteins that bounds to ASOs and interfere in the endosomal escape process and on evaluating their impact in ASO's activity. Overall, the results demonstrated that none of the identified factors play a pivotal role in ASOs release from the endosomes and therefore in their pharmacological efficiency[99,215–219]. Nevertheless, the small effects that the investigators observed when the components of the endosomal machinery were depleted showed that they contribute to the ASO's escape from the endosomes which gave new mechanistic insights of the process. To date, many approaches have been studied to enhance the endosomal escape but none of them were fulfilling all the parameters for clinical use. The most efficient strategies involve the complete endosome compartments rupture which is highly toxic. The complexity of the trafficking machinery, involving many protein and non-protein factors, restrains the successes in overcoming the endosomal trapping of NA-therapeutics. However, the identification of a plethora of potential new targets that can be manipulated, offers many opportunities to improve NA-based therapeutics pharmacological effectiveness.

## A path toward custom-made genetic medicine: the N-of-1 drug's ethical and cost issues

NA-based therapeutics are customizable tools, they thus open the path toward individualized therapies[220]. In 2018, an ASO-based drug was developed by Dr Tim Yu to treat Mila (Milasen), a child suffering from a one-of-a-kind genetic disorder. As Mila's disorder was life threatening, the Milasen benefited from an accelerated FDA authorization process and was administered less than 1 year after the first contact with the patient[221]. Indeed, Mila received the treatment while the animal safety study was still ongoing. The treatment was authorized by the FDA under an expanded-access investigational clinical protocol, which allows "a patient with a serious or immediately life-threatening disease or condition to gain access to an investigational medical product (drug, biologic, or medical device) for treatment outside of clinical trials when no comparable or satisfactory alternative therapy options are available" (Supplementary Fig. 2). After Milasen, further customized ASO drugs for other devastating rare genetic diseases have followed, uncovering unexpected adverse events of the treatments. The emergence of these patient-customized treatments based on NA therapeutics, raises ethical and societal issues in term of regulatory approval and special guidance to govern the development of N-of-1 therapy. In 2019, the publication of the Milasen data was the subject of vigorous debate about where the line should be drawn between letting a child suffer from a fatal disease and trying a potentially harmful experimental treatment. In 2021, the FDA has issued the first ever set of guidances and the first international hub for individualized medicines for rare disease, N = 1 Collaborative (N1C- https://www.n1collaborative.org/) was launched. N1C aims to connect ASO investigators around the world for data sharing and guidelines publications[222], the organization efforts will be extended to other customizable platform technologies such as siRNAs and CRISPR. The emergence of N-of-1 therapies also raises financial issues, sustainable funding for such interventions is challenging. NA-based technology holds the potential to change the therapeutic landscape for many rare diseases in the next ten years. We understand many aspects of the technology and genetic disorders are simpler to treat compared to complex multifactorial pathologies like cancer. This is the time to make rare disease treatments accessible to all. Great efforts have been made recently to raise sufficient funds to enhance the number of patients that can be treated. In 2020, the n-Lorem foundation was founded by Dr Stanley Crooke (Ionis Pharmaceuticals founder) to provide optimal experimental ASO-based therapies to patients with rare diseases. N-Lorem financial supports rely on corporate and disease advocacy organization donations. Rare disease drug discovery is also supported by families or public fundings. Recently, Mila's mother Julia Vitarello co-founded EveryOne Medicines with the support of GV and Khosla Ventures. More recently, the 1 Mutation 1 Medicine (1M1M) European initiative was launched (https://www.1mutation1medicine.eu/). This collaboration aims to establish a scalable European platform for the development and implementation of mutation-specific antisense oligonucleotide treatments for individuals with rare neurological diseases. However, the fundings are still lacking and whether the development of personalized therapies to treat rare diseases will be financially sustainable remains to be seen.

## Conclusion

NA-based drugs such as gene-replacement therapies, RNA therapeutics, or CRISPR gene editing have shown great potential in biomedical applications. Although this type of treatments is not yet a standard therapy, the hopes it raises for patients with rare genetic diseases are immense. Yet, several limitations restrain their widespread use. First, NA-based therapies do not penetrate the BBB to a significant extent which is a serious limitation for the treatment of neurodegenerative diseases. Second, contrary to RNA therapeutics that have transient effect, CRISPR/Cas based treatments can't be reversed if adverse events appear. Strategies to contain the gene editing therapies to specific target tissue; or to restrain the activity of CRISPR gene editing by including on and off switches to prevent the prolonged generation of DNA breaks must thus be developed. Third, the endosomal escape problem is a key aspect of NA-based compounds therapeutic utilization and pharmacological efficiency. Although many strategies to enhance endosomal escape with acceptable toxicity are currently investigated, in expert opinions, we are at the very beginning of the road that leads to an efficient endosomal escape.

Despite the above-mentioned limitations, NA-based therapies provide a unique opportunity for patients suffering from life threatening diseases to

receive tailor-made treatments as demonstrated by the current 19 FDA and/or EMA approved NA therapeutics for the treatment of diverse rare diseases. These therapies will revolutionize the therapeutic landscape for diverse human pathologies provided that sufficient funds are raised to meet patient demands.

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

## Acknowledgements
The research in the author's laboratory was supported by INSERM and grants from the La Ligue (R22030AA), Association pour la Recherche sur les Tumeurs de la Prostate (ARTP) and La Fondation A*Midex -2021 AAP Interdisciplinarité. V.B's postdoc fellowship was funded by la Fondation de France (0131140/WB-2022-44423) and T.K.L.'s postdoc fellowship was funded by ITMO Cancer (C21033AS).

## Author contributions
V.B. composed, edited and finalized the manuscript; T.K.L. and L.C. composed and edited the manuscript and P.R. and D.T. edited the manuscript.

## Competing interests
P. Rocchi, D. Taieb and L. Colleaux are cofounders of SilonTx (https://silontx.com), a biotech company started in April 2024 focusing on precision medicine and nucleic acid therapeutics. The other authors declare no competing interests.
