## [Peer Review File · Communications Biology]

Reviewers' comments:

Reviewer #1 (Remarks to the Author):

Thank you for the review. It is well written and covered broadly in the gene therapy field. The following is a list of comments in order to provide the most up to date information and to be comprehensive or precise on the subject of the review.

1. Line 118-123:

Please add the US regulatory status in your review. The info is as follows and included in the weblink for the authors to extract info for the review: the U.S. Food and Drug Administration approved Zynteglo (betibeglogene autotemcel), the first cell-based gene therapy for the treatment of adult and pediatric patients with beta-thalassemia who require regular red blood cell transfusions. Aug 17, 2022 <https://www.fda.gov/news-events/press-announcements/fda-approves-first-cell-based-gene-therapy-treat-adult-and-pediatric-patients-beta-thalassemia-who#:~:text=Today%2C the U.S. Food and,regular red blood cell transfusions.>

Please also include the effective of Zynteglo, such as the info below and study the weblink: In two phase 3 studies, about 9 out of 10 (89%; 32/36) patients treated with ZYNTGLO stopped transfusions and had a median total hemoglobin level of 11.5 g/dL . Aug 17, 2022

<https://www.zynteglo.com/faqs#:~:text=In two phase 3 studies,level of 11.5 g/dL .>

2. If the approved gene therapy drugs have published clinical trial data, please include them and add in the reference, and please include efficacy info? an example of this is Line 136, but not limited to this line. Please check all commercial products included in the review.

3. Line 155, How many patients are effective... ? The authors need to include a more quantitative summary from literature.

4. Line 160, the authors indicated there are hundreds clinical trials without citing the source on the statement. For a review, all reference data resources need to be cited.

5. Line 188-189, 3 approved drugs in the same class were included, how about their efficacy and safety? Anyway to differentiate them? It would be helpful to search and include those info if they are out there or the authors could study each drug's package insert and summarize the characteristics of each drug.

6. Line 512 Viral-based delivery. The section needs some additional work. The authors may want to consider creating a table to include the vectors of each marketed drug included in the review for clarity and the relevance to the subject of review. The viral-vector topic is reviewed extensively and an updated review is almost out every year. Therefore, there is probably no need to include those references

published before 2021, unless authors clarify more about the purpose of those references to the contents in the review.

7. 568 Non-viral-based delivery. This section needs a major cut. There are very few drugs that are approved using a non-viral vector. If none of the drugs included in your review used a non-viral vector, this section is not essential to the title of your review. It may be included in the future aspects. Additionally, the non-viral based delivery topic has been extensively reviewed recently and any references before 2021 should be excluded unless the authors connect with the content of this review. Listing all the different non-viral-vector types was redundant from previous reviews, the authors need to connect each of those vectors to the current review on rare diseases, or it need to be cut, so that readers can focus on the rare disease subject.

Overall, the authors did a good job compiling the current gene therapy content. But, additional work need to be done to edit and adhere to the title of the review and connect with the core of the review in each section. Please also condense the material to make it up to date on literatures haven't been review yet, and be clear to the readers on the purpose of the content instead of piling up information.

Reviewer #2 (Remarks to the Author):

The manuscript is an informative review, and it is almost up to date once the suggestions are included. I do not have any concern with the paper publishing after addressing the comments. The referencing is inconsistent. For some paragraphs, multiple references were cited but in some, there is none. The description of a gene and involvement in the disease is not consistently referenced. Ref to support the observation are missing in many pages. The discoveries should be acknowledged if it is not common knowledge. Some trials were cited with refs but some were only mentioned as trial names.

The following are some examples. The authors should check throughout the article.

- Page 5 line 162

Include references

- Page 6 line 185-189

Include references

- Page 8 line 275

55 is not a good ref for the sentence but ref 57 is

- Page 8 line 291

Need ref

- Page 9 line 318-335

There was only two ref. The proof of concept study that exon skipping produce smaller but functional protein must be included.

In addition, please also check the following.

- Page 3 line 90-91 “turning off the disease-causing gene”

I am not sure if turning off gene belongs to Gene replacement therapies

- Page 3 line 105

Please include recent gene therapy for Epidermis bullosa

- Page 6 line 215-219

Could move to the delivery section.

- Page 9 line 317 and Table 3.

Milasen should be added

- Page 10 line 348-381

Indicated the ASO is RNase H dependent

- Page 15

Include HSV vector as recent gene therapy for EB uses HSV

- Page 16

Peptide should be in the delivery section.

- Page 17 line 641

Clinical hold is lifted now

- Page 18-19

Include 1M 1M (1 mutation 1 medicine)

- Page 19

include the limitation of Gene Therapy

Table 3, change all DMD ASO into SSO

Figure 3. panel 3 title, change to Steric hindrance : splice modulation.

Figure 5. The innermost circle is appeared unaligned in the printout.

Reviewer #3 (Remarks to the Author):

The paper discusses the transformative potential of nucleic acid-based therapies, encompassing gene replacement and editing approaches as well as RNA-based strategies, in the treatment of genetically-defined disorders. It highlights the versatility of nucleic acid therapeutics, which can be tailored to restore normal biological functions, halt the production of deleterious proteins, increase the synthesis of specific proteins, or correct mutations affecting mRNA splicing. The authors discuss nucleic acid-based drugs currently approved and in development and the paper underscores the significant progress in

nucleic acid therapies, while acknowledging ongoing challenges such as delivery and safety concerns.

While the manuscript mentions the approval of various therapies, outlined examples lack a lot of the context on the mechanisms utilized by these drugs, in particular the antisense oligonucleotide-based compounds and CRISPR/Cas9 editing. ASOs are defined as heteroduplexes, which they are not, as they are not double stranded.

Approved AAV-based therapies are introduced without much context and the later sections outlining vector-based delivery contains conflicting information. In addition, viral-based and non-viral based deliveries are not mutually exclusive and are currently being developed simultaneously as different approaches based on the disease.

Unfortunately, many of the paragraphs are missing references and the overall flow of the paper is disjointed (specifically, the transition between viral gene therapies and oligo therapies). In addition, many abbreviations are either not defined or defined out of order. The paper, overall, contains a large number of grammatical errors.

The authors did a good job collecting the information about approved gene therapies; however, with the consideration that these therapies have been summarized in other reviews, this manuscript needs a lot of polishing.

Rebuttal letter :

Reviewer #1 (Remarks to the Author):

Thank you for the review. It is well written and covered broadly in the gene therapy field. The following is a list of comments in order to provide the most up to date information and to be comprehensive or precise on the subject of the review.

Authors' responses

We would like to thank the Reviewer for the positive feedbacks on the manuscript and for providing constructive and helpful comments. In response, we have made revisions to the manuscript. Please find our responses to the Reviewer's comments below.

1- Line 118-123: Please add the US regulatory status in your review. The info is as follows and included in the weblink for the authors to extract info for the review: the U.S. Food and Drug Administration approved Zynteglo (betibeglogene autotemcel), the first cell-based gene therapy for the treatment of adult and pediatric patients with beta-thalassemia who require regular red blood cell transfusions. Aug 17, 2022 [https://www.fda.gov/news-events/press-announcements/fda-approves-first-cell-based-gene-therapy-treat-adult-and-pediatric-patients-beta-thalassemia-who#:~:text=Today%2C the U.S. Food and, regular red blood cell transfusions.](https://www.fda.gov/news-events/press-announcements/fda-approves-first-cell-based-gene-therapy-treat-adult-and-pediatric-patients-beta-thalassemia-who#:~:text=Today%2C%20the%20U.S.%20Food%20and%20Drug%20Administration%20approved%20Zynteglo%20(betibeglogene%20autotemcel),%20the%20first%20cell-based%20gene%20therapy%20for%20the%20treatment%20of%20adult%20and%20pediatric%20patients%20with%20beta-thalassemia%20who%20require%20regular%20red%20blood%20cell%20transfusions.)

Please also include the effectiveness of Zynteglo, such as the info below and study the weblink: In two phase 3 studies, about 9 out of 10 (89%; 32/36) patients treated with ZYNTEGLO stopped transfusions and had a median total hemoglobin level of 11.5 g/dL .Aug 17, 2022 [https://www.zynteglo.com/faqs#:~:text=In two phase 3 studies, level of 11.5 g/dL .](https://www.zynteglo.com/faqs#:~:text=In%20two%20phase%203%20studies,%20level%20of%2011.5%20g/dL.)

Thanks to the Reviewer for bringing this to our attention. Accordingly, we have added the US regulatory status and the effectiveness of Zynteglo in the review: “In August 2022, the FDA granted approval to Zynteglo®, marking it as the pioneering cell-based gene therapy for beta-thalassemia patients who need regular red blood cell transfusions. In two phase 3 trials (NCT02906202, NCT03207009) approximately 89% (32/36) of patients receiving Zynteglo® attained transfusion independence, with a median total hemoglobin level of 11.5 g/d” (line 130-134).

2. If the approved gene therapy drugs have published clinical trial data, please include them and add in the reference, and please include efficacy info? an example of this is Line 136, but not limited to this line. Please check all commercial products included in the review.

Thanks to the Reviewer for bringing this to our attention. We have added published clinical trial data for all the commercial products that were mentioned in the manuscript.

3. Line 155, How many patients are effective...? The authors need to include a more quantitative summary from literature.

Thank you for your suggestion, we have added more quantitative information: “Spinal muscular atrophy (SMA) is a devastating autosomal recessive and, typically, childhood-onset neuromuscular disorder caused by loss-of-function mutations in the survival of motor neuron 1 (*SMN1*) gene. *SMN1* mutations result in SMN protein deficiency and cause motor neuron degeneration leading to debilitating and often fatal muscle weakness. In SMA murine models, the administration of a «self-complementary» adenovirus serotype 9-associated virus (scAAV9) vector containing one or more copie(s) of the *SMN* transgene (scAAV9-SMN) showed its ability to cross the blood brain barrier and achieve high levels of neuronal transduction. As AAV-based vector genomes can integrate into chromatin, it has been demonstrated that a single intravenous injection of the scAAV9-SMN Zolgensma® (onasemnogene abeparvovec, Novartis) was sufficient to induce long-lasting therapeutic efficiency. The SMA gene-therapy phase 1 trial named START (NCT02122952) results exceeded all expectations as all 15 SMA infants were alive and free of permanent ventilation, compared with a survival rate of 8% in a historical cohort. Based on these results, Zolgensma® received, in May 2019, the FDA approval as the first-ever systemically delivered AAV gene therapy. Two phase 3 clinical trials STR1VE-US (n=22) and STR1VE-EUROPE (n=33) (NCT03306277, NCT03505099) showed safety and efficacy of commercial-grade Zolgensma®, while a recent study confirmed the widespread biodistribution of vector genomes and transgenes throughout the central nervous system and peripheral organs.” (line 169-186)

4. Line 160, the authors indicated there are hundreds clinical trials without citing the source on the statement. For a review, all reference data resources need to be cited.

Thank you for your valuable feedback, we have added more than 40 references to the review, notably the reference #19: Arabi, F., Mansouri, V. & Ahmadbeigi, N. Gene therapy clinical trials, where do we go? An overview. *Biomed. Pharmacother.* **153**, 113324 (2022)”, as suggested for the sentence: “There are currently hundreds of clinical trials worldwide assessing the therapeutic benefits of gene replacement therapies in many different inherited disorders such as metabolic conditions (Friedreich's Ataxia, Wilson disease), cystic fibrosis or hematological conditions”. (line 188-191)

5. Line 188-189, 3 approved drugs in the same class were included, how about their efficacy and safety? Anyway to differentiate them? It would be helpful to search and include those info if they are out there or the authors could study each drug's package insert and summarize the characteristics of each drug.

Thank you for your suggestion. We now summarized in the revised table 2 the main features of these 3 mentioned siRNA drugs. Moreover, in the next paragraphs, these three drugs have been described with the detailed information regarding the mechanism of actions, chemical features, efficacies, approval status (line 250-line 262).

Table 2: Approved siRNA-based therapies for genetic diseases

Trade Name (Drug Name)	Approval year	Disease	Targeted genes	Vector - Modification	Company
Onpattro™ (Patisiran)	2018	Adult patients with hereditary transthyretin mediated amyloidosis	Transthyretin (TTR)	Lipid nanoparticle	Alnylam
Givlaari™ (Givosiran)	2019	Adult patients with acute hepatic porphyria	aminolevulinate synthase 1 (ALAS1)	RNA - GalNAC-conjugation	Alnylam
Oxlumo™ (Lumasiran)	2020	Adult and pediatric patients with primary hyperoxaluria type 1	Hydroxyacid oxidase 1 (HAO1)	RNA - GalNAC-conjugation	Alnylam

6. Line 512 Viral-based delivery. The section needs some additional work. The authors may want to consider creating a table to include the vectors of each marketed drug included in the review for clarity and the relevance to the subject of review. The viral-vector topic is reviewed extensively and an updated review is almost out every year. Therefore, there is probably no need to include those references published before 2021, unless authors clarify more about the purpose of those references to the contents in the review.

We thank the reviewer for the suggestion, the “key issue of delivery strategies” section has been entirely revised as follows:

“One of the great challenges to extend the clinical use of NA-based drugs includes the development of carriers that would effectively deliver them to the target tissues, cells, and organelles. Therapeutics based on nucleic acids are susceptible to enzymatic degradation during their journey in the systemic circulation. Indeed, regardless of the route of administration, naked nucleic acids are rapidly cleared from the organism due to the immune system, and their degradation by endogenous nucleases. In addition, nucleic acids are negatively charged and hydrophilic which impede their passive diffusion across the plasma membranes. As a result, nucleic acids require delivery systems development to protect them from degradation and to deliver them to the target cells for efficient uptake. The different delivery systems for NA-based treatments are well documented in the literature, including viral- and non-viral-based delivery techniques.

Viral-based vectors received huge attention and 80% of gene therapies that are approved and in clinical trials are using retroviruses (including lentiviruses), Adeno-Associated Viruses (AAVs), Adenoviruses (Ads), or Herpes Simplex Viruses (HSVs) to deliver the genetic material (**Figure 5**). The selection of the vector depends on several factors such as its ability to enter the target cells efficiently, its capacity to transfer the genetic material into the nucleus, the size of the genetic material to be delivered, its safety (level of pathogenicity) and immunogenicity.

Ads and AAVs are typically used for *in vivo* delivery consisting in direct administration of the drugs to the patients, the genetic material will remain in episome, this delivery strategy is mainly used

for vaccines against infections and cancer treatments. To treat patients with RD, AAV-based vectors are of the most commonly used viral vectors to deliver gene replacement therapeutics because of their long-term and effective expression of the transgenes, their low immunogenicity and their excellent safety profile in both pre-clinical and clinical investigations. More than 200 AAV-based NA therapeutics are currently being investigated in clinical trials, and as previously mentioned, 3 AAV-based gene replacement therapies reached the commercial market: Glybera, Luxturna, and Zolgensma.

Retroviruses are more applicable for *ex vivo* gene therapy approaches where the foreign genetic material is delivered into hematopoietic cells or other types of stem cells and integrated into the host genome which is advantageous for long-term transgene expression. However, the random integration of the transgenes when using retrovirus-based vectors is a major concern and leukemia development has been reported in clinical assays on X-linked SCID patients. In the clinical trial investigating Strimvelis efficacy in the treatment of ADA-SCID patients, on 50 tested patients only 1 case of leukemia has been reported leading to its approval in 2016. LVs, which belong to the retroviruses family, can incorporate genetic material into the host chromosomes in semi-random manner and display low toxicity. Unlike simple retroviruses, LV vectors could enter in both dividing and non-dividing cells. Due their improved biosafety, a numerous of clinical trials based on LVs-based vectors has been implemented for the treatments of various diseases including genetic disease such as β -thalassemia, cerebral adrenoleukodystrophy, metachromatic leukody. The gene therapy Zynteglo, using LV-based vectors for the transgene delivery, was approved in 2019. The recently approved Lyfgenia™ (lovo-cel), also works by using a LV vector to increase the levels of normal hemoglobin in patients with SCD (**Table 1**).

The extensive understanding of Herpes simplex virus (HSV) sequences and advancements in molecular techniques have paved the way for utilizing HSV as a versatile tool in various applications related to human health. Herpes simplex virus 1 (HSV-1) possesses several characteristics that make it an attractive candidate for therapeutic gene delivery including: episomal delivery, a broad tissue tropism, high transduction efficiency, large transgene capacity and the ability to resist immune clearance via the inhibition of innate and adaptive anti-viral immunity¹⁷⁰. Attenuated or defective HSV1-based¹⁷⁰ vectors have thus been developed and shown promising results in treating various peripheral and central nervous system diseases¹⁸⁴. Yet, this approach has so far only been validated clinically, in 2023, in dermatology for the treatment of rare blistering skin disorder dystrophic epidermolysis bullosa^{29,33}. Vyjuvek™ (beremagene geperpavec) is the first FDA approved HSV1-based gene therapy³¹. The advantages and disadvantages of each of the mainly used viral vectors for NA therapeutics delivery, are summarized in the **Table 6.**" (line 566-621).

We also added which viral vector was used for the approved NA therapeutics in **Table 1**:

Trade Name (Drug Name)	Strategy	Approval year	Disease	Targeted genes	Vector	Company
Glybera (alipogene tiparvovec)	Ex vivo	2012 ^{†2017}	Familial lipoprotein lipase deficiency	lipoprotein lipase (LPL S447X)	Adeno Associate Viru 1 (AAV1)	uniQure

Strimvelis (GSK2696273)	Ex vivo	2016	Severe combined immunodeficiency due to adenosine deaminase deficiency	Adenosine deaminase (ADA)	Gamma-retrovirus	GSK
Luxturna™ (Voretigene Neparvovec-rzyl)	In vivo	2017	RPE65-linked Retinal dystrophy	RPE65	Adeno Associate Virus 2 (AAV2)	Spark Therapeutics
Zynteglo™ (Betibeglogene autotemcel)	Ex vivo	2019 ^{suspended}	β-thalassemia	βA-T87Q-globin (modified β-globin gene)	Lentivirus (BB305)	Bluebird bio
Zolgensma™ (Onasemnogene Apeparvovec-xioi)	In vivo	2019	Spinal Muscular atrophy	SMN1	Adeno Associate Virus 9 (AAV9)	Novartis
Vyjuvek™ (beremagene geperpavec)	Ex vivo	2023	epidermolysis bullosa	COL7A1 gene	Herpes Simplex Virus (HSV-1)	Krystal Biotech
Lyfgenia™ (Lovotibeglogene autotemcel)	Ex vivo	2023	Sickle Cell Disease	Hemoglobin (Hb ^{AT87Q})	Lentivirus (BB305)	Bluebird Bio

7. 568 Non-viral-based delivery. This section needs a major cut. There are very few drugs that are approved using a non-viral vector. If none of the drugs included in your review used a non-viral vector, this section is not essential to the title of your review. It may be included in the future aspects. Additionally, the non-viral based delivery topic has been extensively reviewed recently and any references before 2021 should be excluded unless the authors connect with the content of this review. Listing all the different non-viral-vector types was redundant from previous reviews, the authors need to connect each of those vectors to the current review on rare diseases, or it need to be cut, so that readers can focus on the rare disease subject.

Thank you for your valuable feedback. This section has been revised as follows:

“Non-viral vectors are less popular due to their lower delivery efficacy and gene expression duration. However, viral vectors raise some safety concerns related to their viral origin, oncogenic potentiate, and immunogenicity. Most of the RNA therapeutics, contributing for a majority of FDA or EMA approved NA-based therapeutics, are not conjugated with any delivery platforms, as enormous effort has been made in identifying chemical modifications which improve the stability of nucleic acids and prevent their degradation. The structure of the nucleic acids can be modified at different levels: the nucleobase, the carbohydrate, and the phosphodiester linkage. The most employed modifications in approved ASOs include the phosphorothioate (PS) modification in the phosphodiester linkage and the 2'-O-methoxyethyl (2'-O-MOE) in the sugar group. These modifications, that significantly increase the stability, binding affinity, and reduce the immunogenicity of ASOs, were utilized for the development of Kynamro™, Spinraza™, Tegsedi™, and Waylivra™ for the treatment of various genetic diseases. The approved ASO-based

treatments Exondys51™, Vyondys53™, Viltepso™ and Amondys45™ contain the backbone chemical modification phosphorothioamidate morpholino oligomer (PMO) which improve drug efficacy and specificity, nuclease resistance, solubility in water, and reduce production costs (**Table 3**).

Beside these chemical modifications of the internal structures of nucleic acids, molecules of various nature (lipids, sugars...) can be conjugated with NA therapeutics to effectively enhance their stability and cell penetration, and also facilitates their accumulation in specific organs. As mentioned above, three siRNAs linked to GalNAc moiety received green lights of FDA and/or EMA: Givlaari™, Oxlumo™ and Amvuttra™ (**Table 2**). GalNAc conjugates are clinically convenient as they can be self-administered subcutaneously, resulting in rapid absorption, high uptake, and long half-life (**Figure 5**).

With the development of nanotechnologies, novel nanocarriers are gradually replacing the viral vectors. The commonly found nano-vehicles for NA therapies transportation are summarized in **Figure 5** and are based on lipids (liposomes, solid lipid nanoparticles...), polymers (micelles, dendrimers, nanogels...) and inorganic materials including metal and non-metal nanoparticles. The lipid-based nanoparticles (LNPs) utilization was reported to be effective in delivering drugs to the brain due to their ability to cross the blood-brain barrier (BBB). In particular, LNPs have been intensively studied for small molecules and NA-based drugs delivery and recently attracted great attentions due to big success of mRNA-based vaccines against COVID-19. Approved NA therapeutics using LNP vehicles include gene silencing based-siRNA Onpattro™, and mRNA vaccines against COVID-19 (Comirnaty/tozinameran and mRNA-1273)." (line 622-656)

Overall, the authors did a good job compiling the current gene therapy content. But, additional work need to be done to edit and adhere to the title of the review and connect with the core of the review in each section. Please also condense the material to make it up to date on literatures haven't been review yet, and be clear to the readers on the purpose of the content instead of piling up information.

Reviewer #2 (Remarks to the Author):

The manuscript is an informative review, and it is almost up to date once the suggestions are included. I do not have any concern with the paper publishing after addressing the comments.

We thank the reviewer for the positive evaluation of our manuscript and for providing constructive and helpful comments. In response, we have addressed all of the comments. Please find our responses to the Reviewer's comments below.

The referencing is inconsistent. For some paragraphs, multiple references were cited but in some, there is none. The description of a gene and involvement in the disease is not consistently referenced. Ref to support the observation are missing in many pages. The discoveries should be acknowledged if it is not common knowledge. Some trials were cited with refs but some were only mentioned as trial names.

Thank you for your valuable feedback. References have been extensively revised to address all concerns of the Reviewer, more than 40 references were added to the review.

In addition, please also check the following.

- Page 3 line 90-91 “turning off the disease-causing gene”, I am not sure if turning off gene belongs to Gene replacement therapies.

We agree with the reviewer, the sentence has been revised accordingly: “Gene replacement therapy aims to compensate for abnormal or non-functional genes by delivering genetic materials into cells, thus introducing a healthy copy of the gene.” (line 98-99)

- Page 3 line 105
Please include recent gene therapy for Epidermis bullosa

Thanks to the Reviewer for bringing this to our attention. The following text has been added:

“Due to the ease of access to the diseased tissue, gene therapy may be particularly promising for monogenic skin disorders and researchers have been working on gene therapies for epidermolysis bullosa for over a decade. As a result, in May 2023, the FDA approved, *via* its fast-track process, Vyjuvek™ (Beremagene geperpavec-svdt, also known as B-VEC; Krystal Biotech) a topical gel therapy containing a replication defective herpes-simplex virus type 1 (HSV-1) vector encoding COL7A1 transgene for the treatment of recessive dystrophic epidermolysis bullosa (**Table 1**).” (line 138-144).

- Page 6 line 215-219
Could move to the delivery section.

Thanks a lot for the suggestion. This part has been removed from the text to avoid repetitions as it is already mentioned in the delivery section.

- Page 9 line 317 and Table 3.
Milasen should be added

The Milasen therapy is specifically discussed in the “N of 1 drugs” part in the manuscript (line 710-720). We think that adding this drug in page 9 may burden the review.

Table 3 has been revised as requested.

Trade Name (drug name)	Strategy	Approval year	Disease	Vector Modification	-	Targeted sequence	Company
----------	------------------	---------	------------------------	---	----------------------	---------

Vitravene™ (Fomivirsen)	ASO	1998 ⁺²⁰⁰²	Cytomegalovirus retinitis	RNA Phosphorothioate	- CMV IE2	Ionis Pharmaceuticals
Kynamro™ (Mipomersen)	ASO	2013 ⁺²⁰¹⁸	Homozygous familial hypercholesterol emia	RNA Phosphorothioate /2'-O- methoxyethyl	- ApoB-100	Sanofi and Ionis Pharmaceuticals
Exondys51™ (Eteplirsen)	ASO	2016	Duchenne Muscular Dystrophy	RNA Phosphorodiamid ate morpholino	- Exon 51 of DMD	Sarepta Therapeutics
Spinraza™ (Nusinersen)	ASO	2016	Spinal Muscular Atrophy	RNA Phosphorothioate /2'-O- methoxyethylate d	- Exon 7 of SMN2	Biogen
Tegsedi™ (Inotersen)	ASO	2018	Hereditary Transthyretin- related Amyloidosis	RNA Phosphorothioate /2'-O- methoxyethylate d	- Transthyrety n (TTR)	Ionis Pharmaceuticals
Vyondys53™ (Golodirsen)	ASO	2019	Duchenne Muscular Dystrophy	RNA Phosphorodiamid ate morpholino	- Exon 53 of DMD	Sarepta Therapeutics
Waylivra™ (Volanesorsen)	ASO	2019	Adult Familial Chylomicronemi a syndrome	RNA Phosphorothioate /2'-O- methoxyethyl	- ApoC3	Ionis Pharmaceuticals
Viltepso™ (viltolarsen)	SSO	2020	Duchenne Muscular Dystrophy	RNA Phosphorodiamid ate morpholino	- Exon 53 of DMD	Nippon Shinyaku
Amondys45™ (casimersen/ SRP-4045)	ASO	2021	Duchenne Muscular Dystrophy	RNA Phosphorodiamid ate morpholino	- Exon 45 of DMD	Sarepta Therapeutics
Milasen™	ASO	2019	Batten Disease	RNA Phosphorothioate /2'-O- methoxyethyl	- i6.SA cryptic splice- acceptor site in MFSD8 gene	Boston Children Hospital

•Page 10 line 348-381
Indicated the ASO is RNase H dependent

We now add: “These drugs reduce targeted RNA *via* RNase H1-dependent degradation of the RNA/DNA hybrid” (line 429).

•Page 15

Include HSV vector as recent gene therapy for EB uses HSV

Thank you very much for the valuable suggestion. As requested, we included HSV in both main text (line 609-618) and the **Table 1** and **Table 6**

“The extensive understanding of Herpes simplex virus (HSV) sequences and advancements in molecular techniques have paved the way for utilizing HSV as a versatile tool in various applications related to human health. Herpes simplex virus 1 (HSV-1) possesses several characteristics that make it an attractive candidate for therapeutic gene delivery including: episomal delivery, a broad tissue tropism, high transduction efficiency, large transgene capacity and the ability to resist immune clearance via the inhibition of innate and adaptive anti-viral immunity. Attenuated or defective HSV1-based vectors have thus been developed and shown promising results in treating various peripheral and central nervous system diseases. Yet, this approach has so far only been validated clinically, in 2023, in dermatology for the treatment of rare blistering skin disorder dystrophic epidermolysis bullosa. Vyjuvek™ (beremagene geperpavec) is the first FDA approved HSV1-based gene therapy. The advantages and disadvantages of each of the mainly used viral vectors for NA therapeutics delivery, are summarized in the **Table 6.**”

Table 1: Approved gene replacement therapies for genetic diseases

Trade Name (Drug Name)	Strategy	Approval year	Disease	Targeted genes	Vector	Company
Glybera (alipogene tiparvovec)	Ex vivo	2012 ^{†2017}	Familial lipoprotein lipase deficiency	lipoprotein lipase (LPL S447X)	Adeno Associate Viru 1 (AAV1)	uniQure
Strimvelis (GSK2696273)	Ex vivo	2016	Severe combined immunodeficiency due to adenosine deaminase deficiency	Adenosine deaminase (ADA)	Gamma-retrovirus	GSK
Luxturna™ (Voretigene Neparvovec-rzyl)	In vivo	2017	RPE65-linked Retinal dystrophy	RPE65	Adeno Associate Virus 2 (AAV2)	Spark Therapeutics
Zynteglo™ (Betibeglogene autotemcel)	Ex vivo	2019 ^{suspended}	β-thalassemia	βA-T87Q-globin (modified β-globin gene)	Lentivirus (BB305)	Bluebird bio

Zolgensma™ (Onasemnogen e Abeparvovec- xioi)	In vivo	2019	Spinal Muscular atrophy	SMN1	Adeno Associate Virus 9 (AAV9)	Novartis
Vyjuvek™ (beremagene geperpavec)	In vivo	2023	epidermolysis bullosa	COL7A1 gene	Herpes Simplex Virus (HSV-1)	Krystal Biotech
Lyfgenia™ (Lovotibegloge ne autotemcel)	Ex vivo	2023	Sickle Cell Disease	Hemoglobin (Hb ^{AT87Q})	Lentivirus (BB305)	Bluebird Bio

Table 6: Pros and Cons of the most commonly used viral-based vectors: Ads, AAVs, lentivirus and HSV.

	Advantages	Disadvantages
Adenovirus (Ads)	 ✓ Large payload capacity (up to 36Kb) ✓ High transduction efficiency ✓ The viral genome remains in epichromosomal (not integrate into the host chromatins) ✓ Broad potrism (both diving cells and quiescent cells and various cell types) 	 ✓ Pre-existing viral immune response ✓ Strong immunogenicity against the viral capsid proteins and the transgenic proteins
Adeno-Associated Virus (AAV)	 ✓ Non-pathogenic virus, simple genome, and structure ✓ Low immunogenicity (relatively safe) ✓ Bring long-term and effective expression for the therapeutic genes in broad cell types 	 ✓ Small packing capacity (< 5 Kb) ✓ Adaptive immuno-stimulation induced by the viral capsid proteins ✓ Possibility of oncogenic incorporation of AAV genome into host chromatins ✓ High cost
Lentivirus	 ✓ Integrating vector: long-term transgene expression ability for ex-vivo therapy ✓ Transduction into both dividing cells and quiescent cells ✓ Very low immunogenicity 	 ✓ High risk of mutagenesis insertion: generation of chimeric gene fusions made up of the proviral and host sequences; induction of splicing and production of aberrant transcripts
Herpes Simplex Virus (HSV)	 ✓ Episomal delivery ✓ Broad tissue tropism, ✓ High transduction efficiency, ✓ Large transgene capacity 	 ✓ So far, only been validated clinically in dermatology ✓ Strong cytopathogenicity

✓ Ability to resist immune clearance via the inhibition of innate and adaptive anti-viral immunity	
---	--

•Page 16

Peptide should be in the delivery section.

Thank you for the suggestion. Peptide drug conjugate systems were not described in this review for two reasons: First, although there have been a large number of reports on peptide-based gene delivery systems, most of them are still in the stage of theoretical research and animal experiments, and there are still many challenges before peptide vectors being considered for clinical use; second, as far as we know, they are so far only developed in the field of oncology.

•Page 17 line 641

Clinical hold is lifted now

We thank the reviewer for bringing this to our attention and we have therefore updated the text:

“Despite these encouraging pre-clinical results, the FDA placed, in June 2022, a clinical hold on the phase 2 clinical assay MOMENTUM (NCT04004065) evaluating an exon 51-skipping PPMO, SRP-5051 (Vesleteplirsen) on patients with DMD, after a patient experienced a serious adverse event with grade 3 hypomagnesemia, grade 4 potassium deficiency, muscular cramps, and mild-to-moderate tingling of the extremities. The FDA requested information on all cases of hypomagnesemia, including a small number of nonserious grade 2 cases. The clinical hold was lifted in September 2022 after information was provided by the Company to assess the adequacy of the risk mitigation and safety monitoring plan. Participant enrollment then resumed; the enrollment phase was completed in December 2022.” (683-692)

•Page 18-19

Include 1M 1M (1 mutation 1 medicine)

As suggested by the reviewer we added the following text:

“More recently, the 1 Mutation 1 Medicine (1M1M) European initiative was launched (<https://www.1mutation1medicine.eu/>). This collaboration aims to establish a scalable European platform for the development and implementation of mutation-specific antisense oligonucleotide (ASO) treatments for individuals with rare neurological diseases.” (Line 754-757)

•Page 19 include the limitation of Gene Therapy?

Thanks a lot for the suggestion. We mentioned some limitations of RNA therapeutics in the conclusion section (line 766-776).

Table 3, change all DMD ASO into SSO

Figure 3. panel 3 title, change to Steric hindrance: splice modulation.
Figure 5. The innermost circle is appeared unaligned in the printout.

Thanks a lot for the valuable comment. All the tables and figures revisions have been done.

Reviewer #3 (Remarks to the Author):

The paper discusses the transformative potential of nucleic acid-based therapies, encompassing gene replacement and editing approaches as well as RNA-based strategies, in the treatment of genetically defined disorders. It highlights the versatility of nucleic acid therapeutics, which can be tailored to restore normal biological functions, halt the production of deleterious proteins, increase the synthesis of specific proteins, or correct mutations affecting mRNA splicing. The authors discuss nucleic acid-based drugs currently approved and in development and the paper underscores the significant progress in nucleic acid therapies, while acknowledging ongoing challenges such as delivery and safety concerns.

While the manuscript mentions the approval of various therapies, outlined examples lack a lot of the context on the mechanisms utilized by these drugs, in particular the antisense oligonucleotide-based compounds and CRISPR/Cas9 editing. ASOs are defined as heteroduplexes, which they are not, as they are not double stranded.

We thank the reviewer for this comment, and we agree that the definition of ASOs was not clear. We now changed the sentence in the manuscript to “ASOs are short, synthetic, single-stranded oligodeoxynucleotides that can alter RNA and reduce, restore, or modify protein translation.” (line 330-331)

Approved AAV-based therapies are introduced without much context and the later sections outlining vector-based delivery contains conflicting information. In addition, viral-based and non-viral based deliveries are not mutually exclusive and are currently being developed simultaneously as different approaches based on the disease.

Unfortunately, many of the paragraphs are missing references and the overall flow of the paper is disjointed (specifically, the transition between viral gene therapies and oligo therapies). In addition, many abbreviations are either not defined or defined out of order. The paper, overall, contains a large number of grammatical errors.

We agree with the reviewer that additional references were needed, and we added more than 40 references in the manuscript. The text has been entirely revised and all the abbreviations are now spelled out the first time that we are using them.

The authors did a good job collecting the information about approved gene therapies; however,

with the consideration that these therapies have been summarized in other reviews, this manuscript needs a lot of polishing.

We would like to thank the Reviewer for the valuable comment. The manuscript has been revised intensively, including adding published clinical trial data for all the commercial products that were mentioned in the manuscript. Notably, the “key issue of delivery strategies” section has been entirely revised (line 566-line 656). In detail, we focus more on description of delivery strategies that have been used for approval RNA therapeutics combating genetic diseases. We also provided information about the NA-based therapies that have been recently approved such as Lyfgenia and Casgevy.

REVIEWERS' COMMENTS:

Reviewer #1 (Remarks to the Author):

Thanks for the revision. Nicely done. Appreciate the efforts! The review is fine to be published in this version.

Reviewer #2 (Remarks to the Author):

The authors have addressed all comments.

Reviewer #3 (Remarks to the Author):

The revised review represents a significant enhancement, showcasing substantial improvements in content, particularly in integrating cutting-edge developments within the field of gene therapies. The enhanced flow not only enhances readability but also ensures a coherent presentation of the latest advancements. Moreover, the comprehensive referencing greatly enriches the manuscript, providing readers with an extensive resource to delve deeper into the subject matter.